# CONFIDENT AND ADAPTIVE GENERATIVE SPEECH RECOGNITION VIA RISK CONTROL

**Amit Damri & Bracha Laufer-Goldshtein**
School of Electrical and Computer Engineering
Tel-Aviv University
Tel-Aviv, Israel
{`amitdamti@mail`, `blaufer@tauex`}`.tau.ac.il`

## ABSTRACT

Automatic Speech Recognition (ASR) systems frequently produce transcription errors due to acoustic variability, which require post-processing correction methods. Recent approaches leverage Large Language Models (LLMs) for generative ASR error correction using N-best hypotheses but rely on fixed set sizes regardless of input complexity and do not provide performance guarantees. We propose an adaptive framework that dynamically determines the optimal number of hypotheses for each input using risk control. This mechanism leverages ASR confidence scores and applies Learn then test (LTT) to control the expected relative word error rate degradation compared to the best achievable performance for a given model and hypothesis set. Experimental results demonstrate that our approach provides theoretical guarantees with high-probability bounds while matching or exceeding fixed-size correction baselines and requiring fewer hypotheses on average, achieving substantial computational savings under diverse acoustic conditions.[1]

## 1 INTRODUCTION

ASR systems convert spoken language into text, enabling a wide array of applications from virtual assistants to transcription services (Kheddar et al., 2024). Over the past decade, deep learning advancements have propelled ASR performance, with models like Wav2Vec (Baevski et al., 2020) and Whisper (Radford et al., 2023) achieving remarkable accuracy on benchmark datasets through self-supervised learning and large-scale training. However, ASR remains challenged by real-world variability, including background noise, speaker accents, dialects, homophones, out-of-vocabulary words, and domain shifts, which often lead to transcription errors that degrade downstream tasks (Schneider et al., 2019).

To mitigate these issues, recent research (Yang et al., 2023; Ma et al., 2025; Mu et al., 2025) has explored integrating LLMs with ASR outputs for post-processing. A prominent approach involves generative error correction (GER), where the LLM receives a fixed-size set of hypotheses, produced by the ASR model, and is asked to provide improved transcriptions (Chen et al., 2023; Hu et al., 2024a). Usually the LLM is fine-tuned on sequences of N-best hypotheses to learn mappings from noisy ASR outputs to ground-truth text, demonstrating noise-robust improvements.

Despite these advances, existing GER methods suffer from key limitations. They predominantly rely on a fixed hypothesis set size across all inputs, applying the same $N$ value irrespective of whether the audio is simple (e.g., clear speech) or complex (e.g., accented or noisy), which can result in inefficient resource use—overloading the LLM with redundant hypotheses for straightforward cases or introducing low-quality hypotheses that may degrade correction performance, as can be seen in Fig. 1(a). Furthermore, these approaches lack statistical guarantees on the expected performance, such as bounding the gap to the oracle (best possible) transcription, leaving uncertainty in their reliability and their practical improvement.

To address these shortcomings, we propose an adaptive framework for hypothesis set construction in LLM-augmented ASR, as illustrated in Fig. 1(b). Instead of a static $N$, we dynamically form

---

[1]Our code is available at: `https://github.com/amitdamritau/adaptive-ger`

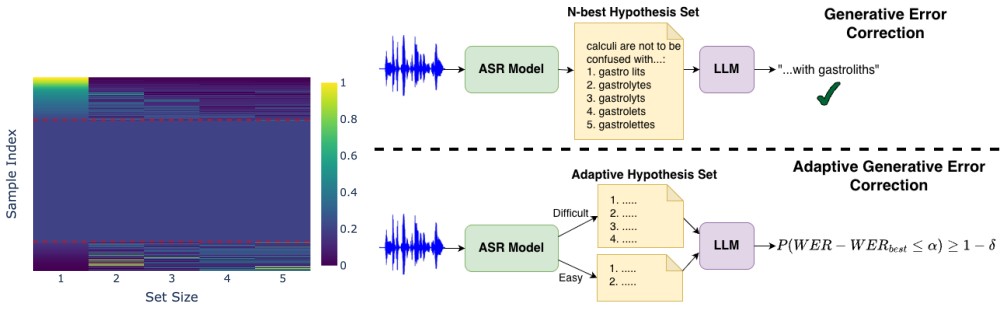

(a) Motivating Example
(b) Method Illustration

Figure 1: **(a)** WER performance patterns across hypothesis set sizes over **TedLium-3**. Samples are grouped by monotonicity: samples that improve with more hypotheses (top), show consistent performance (middle), or degrade with more hypotheses (bottom). **(b)** Comparison of standard GER using fixed 5-hypothesis sets versus our adaptive GER that dynamically selects variable-sized hypothesis sets with risk control to bound relative performance degradation from the oracle.

sets using a threshold rule over the likelihood scores of ASR hypotheses, ensuring only sufficiently plausible candidates are passed to the LLM. We tune these thresholds via Learn then test (LTT) (Angelopoulos et al., 2025), a distribution-free framework that provides risk control with high probability on the expected loss (e.g., word error rate (WER)) relative to the best performance achievable by the model. This adaptive strategy yields smaller average set sizes, reducing computational costs, while empirically achieving comparable or lower WERs compared to fixed-N baselines on diverse benchmarks.

Our main contributions can be summarized as follows:

- We propose an adaptive hypothesis selection framework that leverages ASR confidence scores to dynamically determine the optimal set sizes for each input, replacing the standard fixed-size approach with difficulty-aware resource allocation.

- We introduce the first application of risk control to GER, providing theoretical guarantees alongside empirical validation of effective control over relative performance degradation while enabling principled uncertainty quantification in multi-hypothesis scenarios.

- We demonstrate substantial computational efficiency gains (up to 52% reduction in hypothesis usage) while maintaining correction performance across diverse acoustic conditions, validating both the empirical robustness and practical value of the proposed adaptive selection mechanism.

## 2 RELATED WORK

**Automatic Speech Recognition Error Correction.** Language model rescoring has been extensively employed in ASR systems to enhance recognition accuracy, with external language models reranking N-best hypothesis lists to select optimal transcriptions (Song et al., 2021). Recent advances have moved beyond simple reranking toward generative error correction (GER), where LLMs synthesize improved transcriptions by leveraging complete N-best lists rather than merely selecting among existing candidates (Yang et al., 2023; Radhakrishnan et al., 2023; Yang et al., 2024; Ma et al., 2025; Liu et al., 2025; Ghosh et al., 2024; Mu et al., 2025). Contemporary benchmarks like HyPoradise (Chen et al., 2023) have formalized the hypotheses-to-transcription (H2T) mapping task, enabling systematic evaluation of LLM-based correction methods across diverse acoustic conditions. Our approach builds upon this foundation while introducing reliable and adaptive hypothesis selection via risk control methods.

**Uncertainty Quantification in Language and Speech Processing.** Uncertainty quantification has become critical for deploying natural language processing (NLP) and speech systems in high-stakes applications, with traditional approaches including ensemble methods, Monte Carlo dropout, and calibration techniques for well-calibrated probability estimates (Xiao et al., 2022). Speech processing faces unique challenges due to temporal audio signals and cascading recognition errors, leading to various approaches including acoustic confidence measures and neural uncertainty estimation (Wullach & Chazan, 2023; Rumberg et al., 2025). However, these methods often lack

theoretical guarantees and may not generalize across acoustic conditions, making risk control methods attractive as a principled approach providing distribution-free uncertainty quantification.

**Conformal Prediction and Risk Control Methods.** Conformal prediction (CP) (Vovk et al., 2005; Angelopoulos et al., 2024a) provides a distribution-free and model-agnostic framework for uncertainty quantification that constructs prediction sets with guaranteed coverage under minimal exchangeability assumptions. The framework has found extensive applications across regression, classification, and structured prediction tasks, including recent demonstrations in NLP for machine translation, text classification, and question answering (Campos et al., 2024). Conformal risk control (CRC) extends CP's coverage guarantees to control expected loss functions beyond simple miscoverage, but requires bounded monotone loss functions to maintain distribution-free guarantees (Angelopoulos et al., 2024b). Learn then test (LTT) (Angelopoulos et al., 2025) provides an alternative risk control approach that handles non-monotone loss functions through multiple hypothesis testing with family-wise error rate control, offering high-probability bounds without monotonicity assumptions. Pareto-Testing (Laufer-Goldshtein et al., 2023) provides a computationally and statistically efficient testing strategy for tuning multiple hyperparameters by performing multi-objective optimization prior to hypothesis testing. We leverage both LTT and Pareto-Testing in the development of our reliable and adaptive GER framework.

## 3 PROBLEM FORMULATION

Consider an input audio signal $x \in \mathcal{X}$, and a corresponding transcription $y \in \mathcal{Y}$. Possible transcription *hypotheses* are generated by a pre-trained ASR model using beam search decoding, and the top $N$ are selected:

$$\mathcal{H}_N = \{(\hat{y}_1, c_1), (\hat{y}_2, c_2), \ldots, (\hat{y}_N, c_N)\}, \tag{1}$$

where $\hat{y}_i$ represents the $i$-th hypothesis transcription and $c_i = \log p(y_i|x)$ denotes the log-likelihood score from the ASR model. The hypotheses are ranked by their scores in descending order such that $c_1 \geq c_2 \geq \ldots \geq c_N$, with higher scores indicating higher confidence.

The goal is to learn a mapping function $\mathcal{M}_{\text{H2T}}$ that predicts an improved transcription $\hat{y}^*$ from the N-best list:

$$\hat{y}^* = \mathcal{M}_{\text{H2T}}(\mathcal{H}_N; \theta), \tag{2}$$

where $\theta$ represents learnable parameters. While traditional language model rescoring approaches (Song et al., 2021) re-rank existing hypotheses to select the best candidate, generative error correction (GER)(Ma et al., 2025; Hu et al., 2024a; Yang et al., 2023) represents the current state-of-the-art approach that can synthesize new transcriptions by leveraging information across all N-best hypotheses, potentially producing corrections that do not appear in the original hypothesis list.

LLMs have emerged as powerful tools for this task due to their ability to understand linguistic patterns and perform text generation. The common approaches involve either leveraging existing LLMs with various prompt engineering techniques (Chen et al., 2023; Yang et al., 2023) or fine-tuning a pre-trained LLM to learn the mapping from N-best hypotheses to ground-truth transcriptions (Hu et al., 2024a; Radhakrishnan et al., 2023). The model receives the ranked hypotheses (optionally along with their confidence scores) as input and generates the corrected transcription autoregressively. The training process utilizes pairs $(\mathcal{H}_N, y)$, enabling the model to learn the relationship between ASR error patterns and optimal corrections across diverse acoustic conditions and speaking styles.

However, the conventional approach of using fixed-sized hypothesis sets overlooks a critical observation: not all audio segments require the same number of hypotheses for effective correction. In many cases, a smaller set is sufficient or even preferable for achieving optimal transcription quality, As illustrated in Fig. 1. This phenomenon suggests that additional hypotheses can introduce noise rather than useful signal, motivating the need for adaptive selection mechanisms that can dynamically determine the optimal number of hypotheses based on the specific characteristics of each audio segment.

## 4   BACKGROUND - LEARN THEN TEST FRAMEWORK

Conformal prediction (CP) is a distribution-free framework for uncertainty quantification that relies only on the mild assumption of exchangeability between calibration and test data, without imposing additional assumptions on the underlying model or data-generating process (Angelopoulos et al., 2024a). While standard CP provides guarantees on the miscoverage probability, many practical applications require control of more general risk measures. Conformal risk control (CRC) extends CP by enabling control of the *expected* value of bounded, monotone loss functions (Angelopoulos et al., 2024b). However, the monotonicity requirement restricts its applicability in settings where the loss function exhibits non-monotonic behavior. Further details on both approaches are provided in Appendices D.1 and D.2, respectively.

The Learn then test (LTT) framework addresses these limitations by reformulating risk control as a multiple hypothesis testing problem (Angelopoulos et al., 2025), enabling finite-sample guarantees without monotonicity assumptions. Unlike CRC, which directly bounds the expected risk, LTT provides *high-probability* guarantees: with probability of at least $1 - \delta$ over different draws of the calibration data, the selected $\hat{\lambda}$ satisfies the risk constraint. Consider a calibration dataset $\{(X^{(i)}, Y^{(i)})\}_{i=1}^{m}$, where $X \in \mathcal{X}$ and $Y \in \mathcal{Y}$ denote feature-response pairs. Given a parameterized prediction set function $\Gamma_\lambda : \mathcal{X} \to 2^{\mathcal{Y}}$ and bounded loss function $\ell : 2^{\mathcal{Y}} \times \mathcal{Y} \to [0, B]$, LTT operates on a discrete parameter grid $\Lambda = \{\lambda_1, \lambda_2, \dots, \lambda_k\}$ and associates each $\lambda_j$ with a null hypothesis $H_j : R(\lambda_j) > \alpha$, where $R(\lambda_j) = E[\ell(\Gamma_{\lambda_j}(X), Y)]$ is the expected risk. Rejecting $\mathcal{N}_j$ indicates that $\lambda_j$ achieves the desired risk level. Testing is performed based on valid p-values, which can be computed using the Hoeffding-Bentkus inequality (Appendix A.4) applied to the empirical risk over the calibration set:

$$\hat{R}_m(\lambda_j) = \frac{1}{m} \sum_{i=1}^{m} \ell(\Gamma_{\lambda_j}(X^{(i)}), Y^{(i)}). \tag{3}$$

Since multiple hypotheses are tested simultaneously, it is necessary to control the family-wise error rate (FWER), i.e., the probability of making at least one false rejection. One classical FWER-controlling procedure is the Bonferroni correction, which adjusts the significance level by dividing the threshold $\delta$ by the number of tested hypotheses. However, this global adjustment is often conservative and may substantially reduce statistical power. An alternative approach is to apply fixed sequence testing (FST), which exploits any known natural ordering of $\Lambda$, for instance from low to high risk, by testing the hypotheses sequentially. The procedure proceeds along this ordering and stops at the first non-rejection. In this way, it controls the FWER while typically achieving greater statistical power than the Bonferroni correction.

The LTT guarantees are formally stated in the following theorem.

**Theorem 1** (LTT Finite-Sample Guarantee (Angelopoulos et al., 2025)). *Let $p_j$ be a valid p-value for each null hypothesis $\mathcal{N}_j : R(\lambda_j) > \alpha$, i.e., $P(p_j \leq u) \leq u$ under $\mathcal{N}_j$ for all $u \in [0, 1]$. Then for any FWER-controlling algorithm $\mathcal{A}$ at level $\delta$, the rejection set $\hat{\Lambda} = \mathcal{A}(p_1, \dots, p_k)$ satisfies:*

$$P\left(\sup_{\lambda \in \hat{\Lambda}}\{R(\lambda)\} \leq \alpha\right) \geq 1 - \delta, \tag{4}$$

*where the supremum over an empty set is defined as $-\infty$.*

This guarantee holds without requiring monotonicity assumptions, making LTT particularly well suited to our setting, where the loss function may exhibit complex and non-monotone behavior with respect to $\lambda$. Nevertheless, the FST procedure is still effective in practice, since monotonicity is satisfied in most cases.

## 5   METHOD

### 5.1   ADAPTIVE HYPOTHESIS SELECTION VIA RISK CONTROL

Building on the GER framework, presented in § 3, we propose an adaptive selection mechanism that dynamically estimates the optimal number of hypotheses for each input sample. Rather than

using a fixed set of size $N$, our approach selects the smallest set satisfying the risk constraint, thereby reducing computational cost.

**Adaptive hypothesis set.** We formulate an adaptive hypothesis selection problem within the LTT framework, established in § D.2.1. We define adaptive hypothesis sets, parametrized by $\lambda$ as:

$$\Gamma_\lambda(\mathcal{H}_N) = \{(\hat{y}_1, c_1), \dots, (\hat{y}_n, c_n)\}, \tag{5}$$

where $n$ is the adaptive set size determined according to $\lambda$:

$$n = \min\left\{ j : \sum_{i=1}^{j} s_i \geq \lambda \right\}, \tag{6}$$

and $\mathbf{s} = (s_1, \dots, s_N)$ represents normalized confidence scores derived from ASR log-likelihood scores $\mathbf{c} = (c_1, \dots, c_N)$. The enhanced pipeline becomes $\hat{y}^* = \mathcal{M}_{\text{H2T}}(\Gamma_{\hat{\lambda}}(\mathcal{H}_N); \theta)$, where $\hat{\lambda}$ is the calibrated threshold for controlling expected performance degradation. This approach maintains compatibility with any pre-trained H2T model while reducing computational overhead through principled uncertainty quantification.

**Risk function and LTT.** Our loss is defined with respect to word error rate (WER), which is a standard metric used for evaluating ASR performance. The WER quantifies transcription accuracy by measuring the minimum number of word-level edits required to transform the predicted transcription into the ground truth:

$$\text{WER}(\hat{y}, y) = \frac{S(\hat{y}, y) + D(\hat{y}, y) + I(\hat{y}, y)}{W(y)}, \tag{7}$$

where $S(\hat{y}, y)$, $D(\hat{y}, y)$, and $I(\hat{y}, y)$ represent the number of substitutions, deletions, and insertions, respectively, and $W(y)$ is the total number of words in the reference transcription.

Rather than controlling absolute WER, which requires domain-specific thresholds, we control the *per-sample* relative degradation from the best achievable performance with fixed-set sizes up to $N$:

$$\ell(\Gamma_\lambda(\mathcal{H}_N), y) = \text{WER}(\mathcal{M}_{\text{H2T}}(\Gamma_\lambda(\mathcal{H}_N)), y) - \min_{j \in [N]} \text{WER}(\mathcal{M}_{\text{H2T}}(\mathcal{H}_j), y), \tag{8}$$

where $\mathcal{H}_j = \{(\hat{y}_1, c_1), \dots, (\hat{y}_j, c_j)\}$ denotes the top-$j$ hypothesis set.

This loss function exhibits predominantly monotonic behavior, where enlarging the hypothesis set typically does not worsen performance. Our adaptive selection can identify cases where smaller sets are sufficient or even beneficial. In the worst-case scenario, selecting all $N$ hypotheses converges to the standard fixed-$N$ baseline performance, ensuring no performance degradation from existing methods. Finally, our risk control objective follows the LTT framework:

$$P\left(E[\ell(\Gamma_{\hat{\lambda}}(\mathcal{H}_N), Y)] \leq \alpha\right) \geq 1 - \delta, \tag{9}$$

where $\hat{\lambda}$ is selected from the rejection set obtained by the LTT procedure, ensuring finite-sample control of the expected performance degradation. Our method is summarized in Algorithm 1.

**Score definition.** The selection mechanism relies on a composite score derived from ASR log-likelihoods, designed to adapt flexibly to varying dataset characteristics:

$$\mathbf{s} = \text{softmax}\left(\frac{\phi_\gamma(\mathbf{c})}{\tau}\right). \tag{10}$$

Here, $\phi_\gamma$ denotes an adaptive normalization function and $\tau$ is a temperature parameter. The function $\phi_\gamma$ interpolates between two transformation regimes through a single parameter $\gamma$, enabling the score to adjust to dataset-specific speech quality. To prevent redundancy, penalties are applied when the ASR system generates repeated hypotheses. Further details on the adaptive normalization, design rationale, and repetition handling are provided in the Appendix A.1.

**Score Agnostic.** It is important to note that our method is independent of the specific choice of the score used to define the adaptive set. Previous research in ASR has shown that likelihood values do not always provide a reliable measure of confidence (Li et al., 2021; Ravi et al., 2024). While

---

**Algorithm 1** Learn then Test Selection Procedure

---

**Require:** Calibration set $\{(\mathcal{H}_N^{(i)}, y^{(i)})\}_{i=1}^m$, parameter grid $\Lambda = \{\lambda_1, \lambda_2, \ldots, \lambda_k\}$ ordered from most to least conservative (i.e., $\lambda_1 \geq \lambda_2 \geq \cdots \geq \lambda_k$), error level $\delta$.
 1: **for** $j = 1$ **to** $k$ **do**
 2:     Compute empirical risk $\hat{R}_m(\lambda_j) = \frac{1}{m} \sum_{i=1}^m \ell(\Gamma_{\lambda_j}(\mathcal{H}_N^{(i)}), y^{(i)})$
 3:     Calculate p-value $p_j$ using Hoeffding-Bentkus inequality
 4:     **if** $p_j > \delta$ **then**
 5:         **if** $j = 1$ **then**
 6:             **return** failure (no valid $\lambda$ found)
 7:         **end if**
 8:         **return** $\hat{\lambda} = \lambda_{j-1}$
 9:     **end if**
10: **end for**
11: **return** $\hat{\lambda} = \lambda_k$

---

Li et al. (2021); Ravi et al. (2024) focus on top-label calibration, approaches such as that of Popordanoska et al. (2022) offer canonical calibration, which enables the generation of confidence scores for several top hypotheses that can be seamlessly integrated with our framework. Nevertheless, for simplicity, we demonstrate our method using the more commonly available likelihood values.

**Multi-parameter extension.** While our method optimizes hypothesis set sizes through adaptive thresholds, recent work on multi-objective optimization for risk control suggests a combined optimization and testing approach in the case of multiple parameters that influence multiple objective functions. Pareto Testing (Laufer-Goldshtein et al., 2023) provides an efficient framework for simultaneously calibrating multiple hyperparameters while maintaining statistical guarantees on constrained objectives and optimizing additional free objective function (e.g., set size). In our context, this could enable joint selection of the normalization parameter $\gamma$, temperature $\tau$, and threshold $\lambda$, potentially discovering better performance-efficiency trade-offs than our current approach of fixing $\gamma$ and $\tau$ based on dataset characteristics. Importantly, this joint optimization approach could enable deployment across diverse acoustic conditions without requiring prior knowledge that currently guides our parameter selection strategy. We explore this extension in Appendix D.3.

## 5.2 THEORETICAL CONSIDERATIONS

While risk control frameworks provide principled methods for uncertainty quantification, our ASR application operates under conditions that require careful consideration of theoretical assumptions. We address these considerations and their practical implications.

**Bounded loss.** Risk control frameworks require bounded loss functions; while WER is theoretically unbounded, relative WER degradation is bounded in practice. We enforce boundedness by clipping the loss at $B = 1.25$, i.e., $\ell(\Gamma_{\hat{\lambda}}(\mathcal{H}_N), y) \leq B$, a threshold above which fewer than $0.1\%$ of validation samples fall. As the test set follows a similar distribution, this approximation introduces negligible bias into the theoretical guarantees.

**Monotonicity.** While some risk control methods, like CRC, require monotone loss functions, the monotonicity violations in our application ($\sim 20\%$ of cases) represent precisely the efficiency opportunities our adaptive method exploits - scenarios where smaller hypothesis sets genuinely outperform larger ones. To maintain theoretical guarantees without monotonicity constraints, we employ the LTT framework, which handles non-monotone losses naturally through sequential hypothesis testing, ensuring our approach provides rigorous finite-sample bounds regardless of monotonicity violations. Nevertheless, non-monotonicity can still affect LTT efficiency in practice: since fixed sequence testing advances from the most conservative $\lambda$ toward less conservative values and stops at the first failure, non-monotone behavior may cause early stopping, yielding larger hypothesis sets than necessary. This does not compromise theoretical guarantees but can reduce computational savings. The Pareto Testing extension (Appendix D.3) further addresses this limitation by ordering hypotheses in a nearly monotonic sequence on the optimization set, mitigating the impact of local non-monotonicity on the search trajectory.

## 6 EXPERIMENTAL SETUP

**Datasets and Benchmarks** We evaluate our approach on three datasets from the HyPoradise benchmark (Chen et al., 2023), spanning different acoustic difficulty levels based on average WER performance:

- **TedLium-3** (Hernandez et al., 2018) (avg. WER $\sim 9.5\%$) contains TED Talk recordings with diverse noise, accents, and topics. Following HyPoradise protocol, we sample 50,000 utterances: 35,500 for training/validation, and 14,500 for calibration/test.

- **CHiME-4** (Vincent et al., 2017) (avg. WER $\sim 11.5\%$) contains far-field noisy recordings across different environments. We use the complete train split (9,600 utterances) for train/validation and test-real split (1,320 utterances) for calibration/test. Data was obtained from RobustGER (Hu et al., 2024a), which provides the required ASR likelihood scores.

- **CommonVoice** (Ardila et al., 2020) (avg. WER $\sim 12.5\%$) contains multilingual recordings from diverse speakers with different accents. We select 50,000 samples from train-en split using 35,000 samples for train/validation, and 15,000 samples for calibration/test.

**ASR Hypothesis Generation.** We employ Whisper models (Radford et al., 2023) for $N$-best hypothesis generation via beam search, removing repetitive utterances and selecting top-5 ($N = 5$) hypotheses by posterior probability. TedLium-3 and CommonVoice use Whisper-base (beam-width is 60, following HyPoradise (Chen et al., 2023)), while CHiME-4 uses Whisper-Large-v2 (beam-width is 50, following RobustGER (Hu et al., 2024a)).

**LLM and Training.** We fine-tune LLaMA-2-7B (Touvron et al., 2023) using LoRA (Hu et al., 2022) for efficient H2T mapping. The model generates corrected transcriptions from N-best inputs via standard next-token prediction. Training details, hyperparameters, and computational requirements are in Appendix C.

**Risk Calibration.** We use the validation data to determine both the target risk levels ($\alpha$) and the dataset-specific score function parameters ($\gamma$ and $\tau$), based on the empirical performance and score discriminability patterns. The selection methodology and theoretical considerations are discussed in Appendix A.

Table 1: LTT WER (%) results with LLaMA-2-7B fine-tuning. Baseline: Whisper's top-1 hypothesis. $O_{llm}$: post-LLM oracle. Subscript percentages denote relative WER change vs. vanilla GER (WER column) and relative size reduction vs. constant $N = 5$ (size column).

| Test Set | Baseline | GER | LTT Method | | $\alpha(\%)$ | $\delta$ | Success Rate | $O_{llm}$ |
| --- | --- | --- | --- | --- | --- | --- | --- | --- |
| | | | Set Size | WER | | | | |
| TedLium-3 | 9.3 | 7.53 | $2.48_{-50.08\%}$ | $7.52_{-0.13\%}$ | 2.3 | 0.10 | 0.94 | 5.58 |
| CHiME-4 | 11.49 | 6.24 | $3.866_{-22.68\%}$ | $6.37_{+2.06\%}$ | 2.7 | 0.25 | 0.98 | 4.71 |
| CommonVoice | 12.44 | 8.32 | $3.29_{-34.2\%}$ | $8.51_{+2.28\%}$ | 1.9 | 0.10 | 0.92 | 6.96 |

### 6.1 EVALUATION

**Experimental Protocol.** To ensure statistical reliability, we perform $T = 30$ independent trials with resampled calibration/test splits, allocating 30-50% of test samples for calibration. We set $\delta$ according to the calibration set size, using smaller values for larger datasets where the Hoeffding-Bentkus inequality yields tighter p-value bounds, and larger values for smaller calibration sets to avoid overly conservative thresholds. For the target risk level $\alpha$, we report only configurations where a valid $\hat{\lambda}$ was found during calibration, i.e., where $\alpha$ is achievable given the model and hypothesis set. For instance, very low $\alpha$ values are typically infeasible since even the most conservative selection (always $N = 5$) incurs some irreducible degradation from the oracle. Final results represent mean values across all valid trials.

**Performance Measurements.** We evaluate our approach using WER as the primary metric, as described in § 5. We employ two complementary WER calculation methodologies. The primary approach performs instance-level computation followed by averaging across samples, directly

corresponding to the defined loss function in Eq. (8). As secondary validation, we compute corpus-level WER through concatenation of all predictions and references , which reduces sensitivity to sample length variability and enables comparison with prior works using corpus-level conventions.

**Risk Control Validation.** Beyond standard WER evaluation, we validate the empirical effectiveness of our LTT framework by tracking the success rate of risk control across independent trials. For each dataset, we report the proportion of trials where the risk constraint $R(\hat{\lambda}) \leq \alpha$ is satisfied, validating the high-probability bound $P(R(\hat{\lambda}) \leq \alpha) \geq 1 - \delta$ empirically.

## 7 RESULTS AND ANALYSIS

### 7.1 MAIN RESULTS: PERFORMANCE, EFFICIENCY, AND RISK CONTROL

**Baseline and GER Performance.** Table 1 presents experimental results across datasets. The baseline performance corresponds to Whisper's top-1 hypothesis, establishing the initial recognition accuracy before post-processing. The GER results demonstrate the effectiveness of the fine-tuned LLaMA-2-7B model when provided with a fixed set of top-5 hypotheses. For reference, we include the oracle bound $O_{llm}$, which represents the best possible performance when the LLM receives the optimal number of hypotheses for each sample (between $1 - 5$).

**Adaptive Selection Results.** The results show that GER achieves substantial improvements over the baseline across all conditions, with gains varying according to dataset difficulty. Our adaptive selection framework demonstrates superior computational efficiency while preserving or enhancing correction quality. Our method dynamically determines the optimal number of hypotheses for each input, as reflected in the average set sizes reported. These results validate the effectiveness of our approach across diverse acoustic conditions. On TedLium-3, the method achieves a 51% reduction in average set size while improving performance. CHiME-4 and CommonVoice demonstrate computational savings of 23% and 34% with a modest performance trade-off of 2% relative increase in WER. Notably, adaptive selection sometimes outperforms the fixed-size baseline, suggesting that including too many hypotheses may introduce noise rather than useful information.

**Empirical Guarantee Verification.** Regarding the reliability of our selection mechanism, success rates consistently exceed the theoretical minimum of $1 - \delta$ across all trials, confirming that the high-probability bound $P(R(\hat{\lambda}) \leq \alpha) \geq 1 - \delta$ is satisfied in practice - a property absent in prior methods and made possible through the LTT framework. The observed gap between empirical success rates and the theoretical minimum is expected, reflecting the conservative nature of finite-sample Hoeffding-Bentkus bounds; as validated by experiments with varying calibration set sizes, this gap narrows as more calibration data becomes available.

**Performance-Compute Trade-off.** Figure 2 illustrates the performance-compute trade-off characteristics of our adaptive approach compared to fixed set sizes across all datasets. Each subplot displays the WER performance curve for constant set sizes $N = 1$ through $N = 5$, with vertical reference line indicating the $O_{llm}$ oracle performance bound. We observe that our method's operating points consistently demonstrate better tradeoffs relative to the fixed-set performance curve, achieving computational efficiency gains while maintaining competitive or even improved error rates.

### 7.2 ANALYSIS

To illustrate the adaptive selection mechanism, Table 2 presents three representative cases, highlighting how score distributions relate to optimal set sizes.

**Case 1: Full Set Required** (Common Voice): When ASR scores exhibit narrow gaps (-0.42 to -0.51), our method correctly identifies the need for comprehensive information. The LLM progressively refines its prediction across set sizes, ultimately achieving perfect accuracy with the complete hypothesis set by correctly generating "gastroliths" rather than the various incorrect alternatives ("gallstones," "gastrolytes"). The compressed score distribution leads our normalization to select larger sets, aligning with the empirical benefit of additional hypotheses.

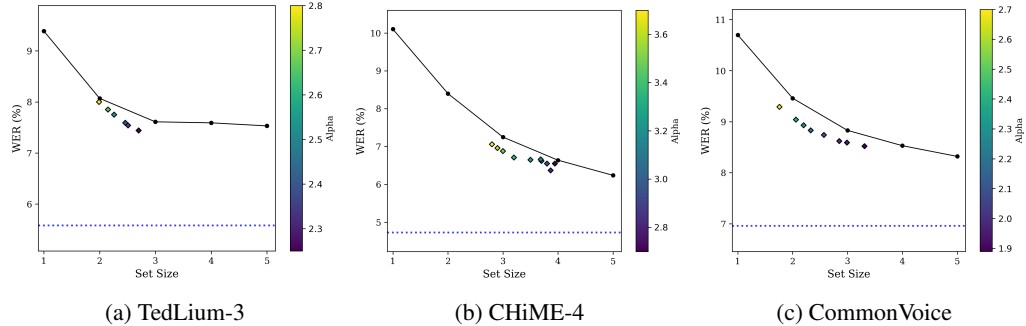

|   (a) TedLium-3   |   (b) CHiME-4   |   (c) CommonVoice   |

Figure 2: Performance-compute trade-offs across datasets. Plots show WER vs. set size for constant set sizes (connected line), oracle performance level (dashed line), and our adaptive method working points (markers), where marker colors encode corresponding $\alpha$ values.

Table 2: Representative examples showing the relationship between ASR score distributions and optimal set sizes. Case 1 demonstrates progressive improvement with larger sets, Case 2 shows degradation beyond the optimal single hypothesis, and Case 3 illustrates performance plateau enabling computational savings.

| Case | Hypotheses | Score | LLM Predictions by Set Size | WER per Size (%) |
|---|---|---|---|---|
| **Case 1: Full Set** | H1: calculi are not to be confused with gastro lits | -0.42 | Size 1: ...with gallstones | 12.5 |
|  | H2: calculi are not to be confused with gastrolytes | -0.44 | Size 2: ...with gastrolytes | 12.5 |
|  | H3: calculi are not to be confused with gastrolyts | -0.47 | Size 3: ...with gastrolytes | 12.5 |
|  | H4: calculi are not to be confused with gastrolets | -0.50 | Size 4: ...with gastrolytes | 12.5 |
|  | H5: calculi are not to be confused with gastrolettes | -0.51 | Size 5: ...with gastroliths | 0.0 |
|  | *GT: calculi are not to be confused with gastroliths* |  |  |  |
| **Case 2: Single Opt.** | H1: ...medical team assign of the ship... | -0.21 | Size 1: ...team a sign of... | 0.0 |
|  | H2: ...medical team a sign of the ship... | -0.31 | Size 2-5: ...team assigned to... | 21 |
|  | H3: ...medical team assigned to the ship... | -0.37 |  | 21 |
|  | H4: ...medical team assigned of the ship... | -0.41 |  | 21 |
|  | H5: ...medical team assigned the ship... | -0.43 |  | 21 |
|  | *GT: ...medical team a sign of the ship...* |  |  |  |
| **Case 3: Plateau** | H1: ...new york state sold about seventy seven point one million of... | -0.46 | All sizes: separately new york | 6.25 |
|  | H2: ...new york state sold about seventy seven point one million in... | -0.47 | state sold about seventy seven | 6.25 |
|  | H3: ...here it states all about seventy seven point one million in... | -0.47 | point one million dollars in | 6.25 |
|  | H4: ...new york state sold about seventy seven point one million of... | -0.49 | certificates of participation | 6.25 |
|  | H5: ...new york state sold about seventy point one seven million dollars in... | -0.49 |  | 6.25 |
|  | *GT: ...seventy seven point one million dollars of...* |  |  |  |

**Case 2: Single Hypothesis Optimal** (TedLium-3): When the top hypothesis achieves perfect accuracy and exhibits substantial score separation (-0.21 vs -0.31), additional hypotheses degrade performance from $0\%$ to $21\%$ WER. In this case, the discriminative score gap correctly signals high confidence in the first hypothesis, leading our method to favor minimal sets. This demonstrates that additional hypotheses can introduce harmful noise.

**Case 3: Performance Plateau** (CHiME-4): When WER remains constant (6.25%) across all set sizes, our method demonstrates computational efficiency potential. While the tight score clustering (-0.46 to -0.49) would typically lead our normalization to select larger sets, this case illustrates where our approach provides a safety net—in the worst case, we select all 5 hypotheses and achieve identical performance to the baseline, but when score normalization successfully identifies the plateau, we achieve the same WER with reduced computational cost.

These examples show that adaptive selection produces smaller sets when scores are highly discriminative and larger sets when scores are compressed. This supports dynamically adjusting set sizes based on the underlying score distribution rather than relying on fixed configurations.

## 7.3 ABLATION STUDIES

We briefly report several ablation studies that we performed to validate different aspects of our proposed framework.

**Alternative Problem Formulations.** Our preliminary experiments evaluate multiple CP and risk control methods including absolute WER targets, coverage-based objectives for samples below specified WER thresholds, and bounded-WER hypothesis guarantees, following approaches from prior ASR uncertainty quantification works (Ernez et al., 2023). These alternatives consistently yielded inferior empirical performance compared to our relative loss, defined in Eq. 8. Absolute

WER targets operate at a global level without instance-specific optimization, while bounded-WER guarantees (Ernez et al., 2023) showed poor correlation between hypothesis quality and final LLM output quality, motivating our relative degradation formulation that adapts to each sample's achievable performance range.

**Training Set Size Analysis**: We examined our choice of training the LLM with constant-5 hypothesis sets, while evaluating with variable set sizes. To this end, we conducted comprehensive ablation experiments training separate LLaMA-2-7B models on fixed set sizes (1-5 hypotheses), as well as dynamic sizes, then evaluating each model across all possible test set sizes. The results are reported in Tab. B.2. This $6 \times 5$ result matrix reveals that while specific combinations (e.g., train-3/test-3) occasionally outperformed the baseline, the constant-5 trained model achieved optimal average WER across all test configurations. These results confirm that our adaptive approach provides genuine improvements over the best achievable fixed-size baseline, establishing the validity of our comparative framework.

**Scalability to Larger Models and Zero-Shot Settings.** To evaluate generalizability beyond our LLaMA-2-7B baseline, we conducted experiments with LLaMA-2-13B (fine-tuned) and GPT-3.5-turbo (zero-shot prompting). Results demonstrate that our framework maintains consistent performance-efficiency trade-offs across both larger model scales and deployment scenarios where fine-tuning is not feasible. The computational savings persist relative to the increased inference costs, confirming that our adaptive selection mechanism provides value across different model architectures and prompting paradigms. This addresses practical deployment considerations where model size and training constraints vary significantly. Detailed results are provided in Appendix B.2.

**Cross-Domain Extension.** We extended our framework to speech translation tasks using the Gen-Translate paradigm on multilingual datasets. with some methodological adaptation. Results show successful transfer with substantial computational savings while maintaining competitive translation quality. This cross-domain validation demonstrates broader applicability beyond ASR to generative error correction scenarios involving N-best hypothesis integration, directly addressing the broader impact potential in the GER community. Full methodology and results in Appendix B.4.

**CRC Implementation.** We also implemented the CRC framework as an alternative risk control method. While CRC lacks theoretical guarantees due to monotonicity violations in our application ($\sim 20\%$ of cases), it achieves similar empirical performance to our LTT approach. This demonstrates that both frameworks effectively exploit the same underlying adaptive selection patterns, with LTT providing the additional benefit of rigorous theoretical validation. The CRC implementation and comparative analysis are detailed in Appendix D.2.

**Multi-Parameter Optimization via Pareto Testing.** We explored an extension of our framework that jointly optimizes all three parameters ($\gamma$, $\tau$, $\lambda$) simultaneously using Pareto Testing (Laufer-Goldshtein et al., 2023), eliminating the need for dataset-specific parameter pre-selection. Results demonstrate consistent performance-efficiency trade-offs with our main LTT approach, while discovering parameter combinations that achieve superior Pareto frontiers compared to fixed-parameter baselines, offering increased adaptability to unseen acoustic conditions. Full methodology and results are provided in Appendix D.3.

## 8 CONCLUSION AND FUTURE WORK

This work presents an adaptive framework for hypothesis selection in generative ASR error correction, addressing computational inefficiency through principled uncertainty quantification. Our method employs LTT to dynamically determine optimal hypothesis set sizes, providing rigorous theoretical guarantees with high-probability bounds while demonstrating substantial computational savings and maintaining competitive performance across datasets with diverse acoustic conditions. Validation across larger language models, zero-shot settings, and cross-domain speech translation tasks confirms the framework's broad applicability and robustness across different deployment scenarios. The framework requires only calibration without model retraining, enabling straightforward adoption in existing systems. Future work could investigate confidence-driven adaptive compute allocation in multi-model systems, including reasoning and agent-based applications, where similar mechanisms for identifying and reducing computational costs may achieve comparable performance with greater efficiency.

ACKNOWLEDGMENTS

This work was supported by the Israel Science Foundation (Grant No. 2094/25), and the Israeli Ministry of Innovation, Science and Technology (Grant No. 1001818518).

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

## A  RISK CONTROL IMPLEMENTATION DETAILS

### A.1  SCORE FUNCTION DESIGN

#### A.1.1  MOTIVATION

Our analysis revealed that score distributions vary significantly across datasets with different noise characteristics. Higher signal-to-noise ratio conditions (e.g., TedLium-3) produce more discriminative ASR confidence scores, while challenging acoustic environments (e.g., CommonVoice) yield compressed score distributions. Temperature-only adaptation proved insufficient, creating overly homogeneous score distributions that degraded selection quality. Consequently, we developed the two-level normalization strategy with parameter $\gamma$, enabling adaptive score transformation based on dataset difficulty while maintaining robustness across parameter choices.

#### A.1.2  FUNCTION DESIGN

The normalization function $\phi_\gamma(c)$ smoothly interpolates between two transformation regimes based on a single parameter $\gamma \in [0, 1]$:

$$\phi_\gamma(\mathbf{c}) = (1 - \gamma) \cdot (-\mathbf{c}^{-1}) + \gamma \cdot \mathbf{c}, \tag{11}$$

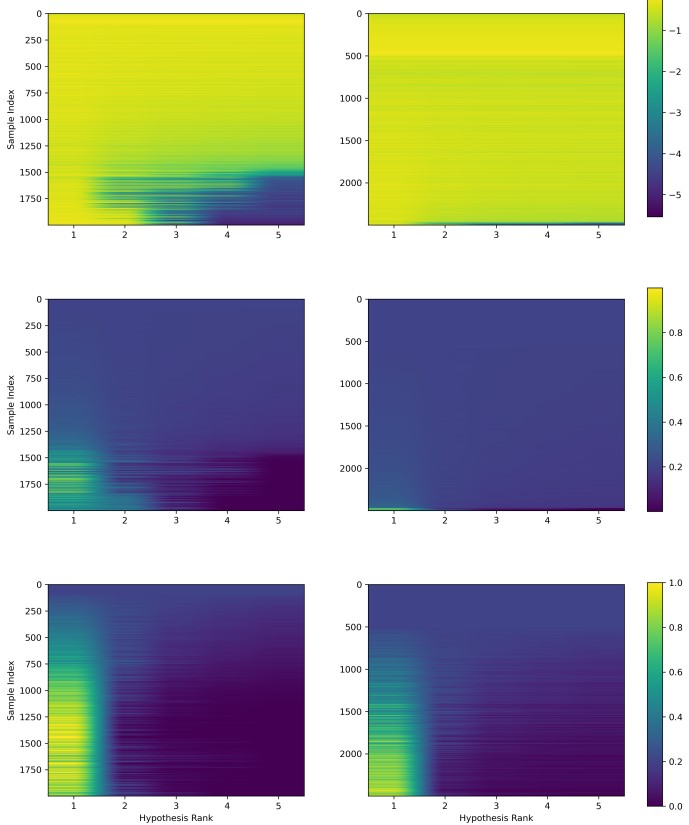

Figure A.1: ASR confidence score distributions for TedLium-3 (left) and CommonVoice (right) across processing stages: raw scores (top), softmax normalization (middle), and full transformation (bottom).

where $\mathbf{c}^{-1}$ denotes element-wise reciprocal operation, i.e., $(\mathbf{c}^{-1})_i = 1/c_i$ for each component $i$.

The identity transformation ($\gamma = 1$) preserves natural ASR score differences for high-SNR conditions, while the reciprocal transformation ($\gamma = 0$) amplifies small differences between compressed scores for challenging acoustic conditions. The parameter $\gamma$ controls smooth transitions between these regimes.

### A.1.3 PARAMETER SELECTION STRATEGY

We base parameter selection on signal-to-noise ratio characteristics and empirical validation.

Figure A.1 illustrates score evolution across processing stages for different acoustic conditions. TedLium-3's high-SNR conditions produce naturally discriminative scores, requiring only sharpening via low temperature ($\tau = 0.05$) while preserving relationships ($\gamma = 1.0$). CommonVoice's challenging conditions with compressed distributions require reciprocal amplification ($\gamma = 0.0$) followed by moderate temperature ($\tau = 1.0$). CHiME-4 represents intermediate conditions requiring balanced transformation ($\gamma = 0.5$).

To validate this SNR-based rationale, we conducted systematic grid testing across the parameter space. Figure A.2 shows success regions where the method achieves valid risk control with better performance-compute trade-offs than fixed baselines.

The heat-maps confirm that successful regions align with our SNR-based parameter selection, with edge cases (high/low SNR) showing concentrated success areas and intermediate conditions requiring parameters within mid-value ranges.

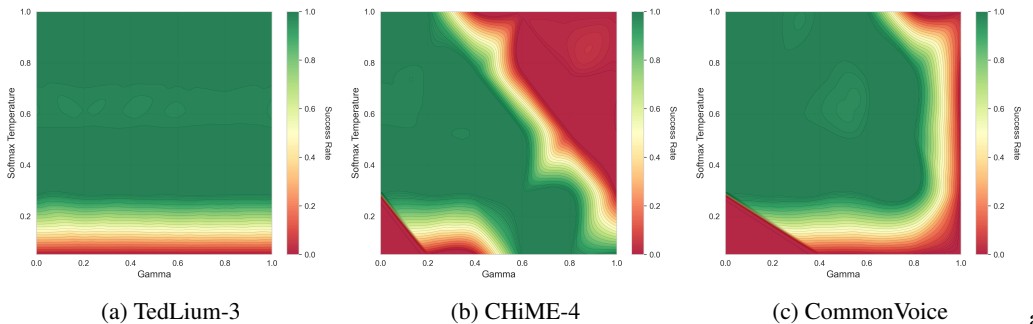

(a) TedLium-3        (b) CHiME-4        (c) CommonVoice    as

Figure A.2: Parameter selection grid test results showing success regions across $(\gamma, \tau)$ parameter space. Success is defined as achieving valid risk control with superior performance-compute trade-offs compared to fixed baselines. Heat-maps confirm that successful regions align with SNR-based parameter selection strategy.

### A.1.4 ROBUSTNESS ANALYSIS

Grid testing across uniformly sampled parameter combinations demonstrates method robustness. Success criteria require: (1) valid $\lambda$ selection with controlled risk, and (2) superior performance-compute trade-offs compared to fixed-set baselines.

Results show overall success rate of 70% (61/88 combinations), with per-dataset rates: TedLium-3: 80% (24/30), CHiME-4: 62% (18/29), CommonVoice: 66% (19/29). This indicates parameter selection is an optimization step rather than a critical requirement, providing practitioners flexibility while maintaining performance guarantees.

### A.1.5 AUTOMATED PARAMETER SELECTION

To automate parameter selection for deployment across unseen acoustic conditions, we investigated the relationship between score distribution characteristics and working point success. Our intuition is that meaningful working points should exhibit score vectors that respond differently across varying input difficulties, with entropy serving as a natural measure of this distributional behavior.

Analysis confirmed that score vector entropy serves as a reliable predictor of parameter effectiveness - successful working points exhibit distinct entropy patterns compared to failed configurations. Based on this observation, we developed a simple entropy-based rule: thresholding score vector entropy at 4.85 effectively discriminates between suitable and unsuitable parameter combinations.

We validated this rule on previously unseen parameter pairs, achieving prediction precision exceeding 80% and F1 scores above 75%. This enables practitioners to assess parameter suitability on new datasets without extensive manual calibration, providing a practical deployment pathway that transforms manual parameter tuning into principled selection with automated validation.

It should be noted that parameter sensitivity increases when targeting tighter performance bounds (lower $\alpha$ and $\delta$ values), as the method operates in narrower feasible regions where precise parameter selection becomes more critical.

### A.1.6 HANDLING HYPOTHESIS REPETITIONS

When the ASR system produces fewer than $N$ unique hypotheses, repeated hypotheses receive exponentially decaying scores to avoid overweighting redundant information:

$$R_{i,r} = s_i \cdot \beta^r, \tag{12}$$

where $R_{i,r}$ is the adjusted score for hypothesis $i$ with repetition count $r$, and $\beta \in (0, 1)$ is the decay factor. This mechanism is based on the assumption that repeated hypotheses provide no additional information for well-calibrated models, which we validate empirically in our experiments.

## A.2 TARGET RISK

### A.2.1 RISK TARGET CALIBRATION

We establish target risk levels based on the performance range achievable through fixed hypothesis set selection. The feasible degradation range spans from fixed-1-hypothesis (worst case) to fixed-5-hypotheses (best case) performance. Since our adaptive method dynamically selects smaller sets with minimal degradation, targeting risk bounds within this empirically-derived range represents the natural operating regime. These empirically-derived ranges are $[1.95, 3.8]$ for TedLium-3, $[1.53, 5.76]$ for CHiME-4, and $[1.38, 3.74]$ for CommonVoice. We uniformly sample target risk values within the validated degradation range. The specific choice within this range determines the desired performance-compute trade-off: values closer to 1-hypothesis' risk prioritize computational efficiency, while values approaching 5-hypotheses' risk emphasize performance preservation. The different risk–control frameworks require different $\alpha$–calibration approaches. For CRC, we directly use validation–derived values as the target expected risk levels, since CRC bounds $\mathbb{E}[R(\hat{\lambda})] \leq \alpha$. On the other hand, LTT and Pareto-testing, provide high–probability bounds $\Pr\left(R(\hat{\lambda}) \leq \alpha\right) \geq 1 - \delta$, meaning that $\alpha$ represents approximately the $(1 - \delta)_{th}$ percentile of the empirical risk distribution rather than its expectation. To achieve equivalent average performance, those frameworks therefore require $\alpha$ values that are higher than the target expected empirical risk. As the calibration set sizes is smaller, the Hoeffding-Bentkus inequality produces more conservative p-values, leading to more conservative set selections with higher empirical coverage. In practice, LTT and Pareto typically yield valid selections for $\alpha$ values from the upper part of the feasible range, while CRC can achieve valid selections across the full range due to its expectation-based formulation. We report results only for $\alpha$ values that demonstrate successful risk control within the validated range, ensuring both theoretical validity and practical utility across different performance-efficiency preferences.

### A.2.2 PERFORMANCE TRADE-OFFS

The target risk $\alpha$ directly controls the performance-compute trade-off by determining acceptable expected relative WER degradation from oracle performance. Lower $\alpha$ values (tighter bounds) require larger hypothesis sets to achieve the target risk level, due to average WER monotonicity on average, while higher $\alpha$ values (looser bounds) enable smaller sets with acceptable performance degradation.

Table A.1: Effect of risk tolerance $\alpha$ on performance-compute trade-offs (CommonVoice dataset).

| Risk Tolerance ($\alpha$) | Avg. Set Size | WER (%) |
|---|---|---|
| 0.21 (tighter) | 3.73 | 8.53 |
| 0.22 | 3.49 | 8.59 |
| 0.23 | 3.14 | 8.66 |
| 0.24 (looser) | 2.99 | 8.73 |

Table A.1 demonstrates this relationship empirically. As $\alpha$ increases, average set sizes decrease, while WER increases as well. The different operating points in Figure 2 correspond to varying $\alpha$ selections within these validated ranges.

## A.3 OTHER PARAMETERS SELECTION

We set additional framework parameters based on validation analysis to ensure robust performance across datasets. The repetition penalty $\beta = 1.25$ handles duplicate hypotheses by applying exponential decay to repeated entries, preventing overweighting of redundant information. The loss bound $B = 1.25$ accounts for rare cases exceeding 100% relative WER degradation, satisfying the bounded loss requirement for theoretical guarantees. These parameters were determined through empirical validation to maintain stability across varying hypothesis quality distributions while preserving the risk control framework's theoretical foundations.

A.4   HOEFFDING-BENTKUS P-VALUE COMPUTATION

For LTT implementation, we compute valid p-values using the Hoeffding-Bentkus inequality. Given empirical risk $\hat{R}_m(\lambda_j)$ on the calibration set, the p-value for hypothesis $\mathcal{N}_j : R(\lambda_j) > \alpha$ is:

$$p_j^{HB} = \min\left\{\exp\{-nh_1(\hat{R}_m(\lambda_j) \wedge \alpha, \alpha)\}, eP\left(\text{Bin}(n, \alpha) \geq \lfloor n\hat{R}_m(\lambda_j)\rfloor\right)\right\}, \quad (13)$$

where $h_1(a, b) = a\log(a/b) + (1 - a)\log((1 - a)/(1 - b))$ and $n$ is the calibration set size. This provides finite-sample valid p-values without distributional assumptions, enabling the sequential testing procedure in Algorithm 1.

# B   ABLATION STUDY

## B.1   CORPUS-LEVEL WER

We compute corpus-level WER through concatenation of all predictions and references using the `evaluate`[2] package. Results are summarized in Table B.1.

Table B.1: Corpus-level WER (%) results with LLaMA-2-7B fine-tuning. Our method results represent one operating point from Figure 2. Results show consistent trends with instance-level averaging (Table D.1) despite different absolute values, demonstrating robustness across evaluation methodologies. Subscript percentages denote relative WER change vs. vanilla GER and relative size reduction vs. constant $N = 5$.

| Test Set | GER | Our Method | | $\mathbf{O}_{llm}$ |
|---|---|---|---|---|
| | | Set Size | WER | |
| TedLium-3 | 5.05 | $2.21_{55.8\%}$ | $5.05_{0.0\%}$ | 3.03 |
| CHiME-4 | 6.37 | $3.8_{24.0\%}$ | $6.6_{+3.6\%}$ | 4.78 |
| CommonVoice | 7.8 | $3.07_{38.6\%}$ | $7.95_{+1.9\%}$ | 6.31 |

## B.2   ANALYSIS OF TRAINING SET SIZE EFFECTS

The ablation results presented here use a simplified experimental setup with different hyperparameters and dataset splits compared to the main experiments; nonetheless, they demonstrate consistent patterns that validate our core findings.

The results reveal several key patterns that validate our experimental design. The constant-5 training approach achieves the lowest average WER (7.79%) across all test configurations, confirming its superiority as a baseline model. While diagonal elements (matching train/test sizes) occasionally show local optima—such as train-3/test-3 achieving 6.58% versus the train-5/test-3 result of 6.74%—these improvements are marginal and inconsistent across the full evaluation matrix.

Models trained on smaller hypothesis sets exhibit clear performance degradation when tested on larger sets, as expected. The train-1 model struggles significantly with multi-hypothesis inputs, achieving 11.65% WER on 5-hypothesis tests compared to 6.38% for the train-5 model, demonstrating the importance of exposure to diverse hypothesis patterns during training.

The dynamic training model, despite having access to variable set sizes during training, underperforms the constant-5 baseline (8.43% vs 7.79% average WER). This degraded performance likely stems from the increased complexity of learning hypothesis-to-transcription mappings across varying input lengths simultaneously, creating a more challenging optimization landscape that prevents the model from fully mastering any single configuration. The model must learn to handle the variability in input structure while maintaining transcription quality, leading to suboptimal specialization compared to the focused constant-5 training regime.

---

[2]https://pypi.org/project/evaluate/

These results establish that our adaptive approach provides genuine improvements over the best achievable fixed-size baseline, validating the comparative framework used throughout our main experiments.

Table B.2: Training Set Size Ablation Study: WER (%) across different training and test configurations on CHiME-4 dataset

| Train \ Test | 1-hyp | 2-hyp | 3-hyp | 4-hyp | 5-hyp | Average |
|---|---|---|---|---|---|---|
| Train-1 | **10.32** | 10.85 | 11.12 | 11.38 | 11.65 | 11.06 |
| Train-2 | 10.72 | **8.55** | 8.92 | 9.15 | 9.41 | 9.35 |
| Train-3 | 10.89 | 8.95 | **6.58** | 6.89 | 7.12 | 8.09 |
| Train-4 | 10.95 | 9.12 | 6.89 | **6.52** | 6.71 | 8.04 |
| Train-5 | 10.48 | 8.69 | 6.74 | 6.64 | **6.38** | **7.79** |
| Dynamic | 11.23 | 9.45 | 7.32 | 7.18 | 6.95 | 8.43 |

## B.3 SCALABILITY TO LARGER LANGUAGE MODELS

To evaluate the generalizability of our adaptive framework beyond the LLaMA-2-7B baseline, we conducted experiments with LLaMA-2-13B (fine-tuned) and GPT-3.5-turbo (zero-shot prompting). These experiments assess whether the computational efficiency gains and adaptive selection benefits persist with more capable and resource-intensive models. All experiments use the LTT calibration procedure with dataset-specific $\alpha$ values, as detailed in Section A.2.

### B.3.1 LLaMA-2-13B RESULTS

We fine-tuned LLaMA-2-13B using identical training procedures, hyperparameters, and evaluation protocols as our main experiments. Table B.3 presents representative working points for each dataset.

Table B.3: LLaMA-2-13B adaptive selection results. Results shown for one representative working point per dataset; multiple valid working points exist across the parameter space.

| Test Set | Baseline | GER | Our Method | | $\alpha(\%)$ | $\delta$ | $O_{llm}$ |
|---|---|---|---|---|---|---|---|
| | | | Set Size | WER | | | |
| TedLium-3 | 8.0 | 6.47 | $2.40_{51.9\%}$ | $6.46_{-0.01\%}$ | 2.1 | 0.25 | 4.69 |
| CHiME-4 | 11.49 | 7.92 | $3.88_{22.4\%}$ | $8.16_{+3.06\%}$ | 1.25 | 0.25 | 7.26 |
| CommonVoice | 14.1 | 8.10 | $3.50_{30.1\%}$ | $8.28_{+2.27\%}$ | 2.0 | 0.25 | 6.50 |

The results demonstrate that our framework maintains its value proposition with larger models: TedLium-3 achieves 51.9% computational savings with identical performance, while CHiME-4 and CommonVoice show modest performance trade-offs (3.06% and 2.27% relative WER increase) for substantial efficiency gains (22.4% and 30.1% hypothesis reduction). These results confirm that the performance of the adaptive selection mechanism generalizes across model scales.

### B.3.2 GPT-3.5-TURBO ZERO-SHOT RESULTS

To assess applicability without fine-tuning, we evaluated GPT-3.5-turbo using zero-shot prompting based on templates from HyPoradise (Chen et al., 2023) and (Ma et al., 2024).

Table B.4: GPT-3.5-turbo zero-shot adaptive selection results. Results shown for one representative working point per dataset.

| Test Set | Baseline | GER | Our Method | | $\alpha(\%)$ | $\delta$ | $O_{llm}$ |
|---|---|---|---|---|---|---|---|
| | | | Set Size | WER | | | |
| CommonVoice | 14.1 | 11.73 | $2.31_{42.3\%}$ | $11.81_{+0.67\%}$ | 1.5 | 0.25 | 10.51 |
| CHiME-4 | 11.49 | 9.77 | $2.19_{56.1\%}$ | $9.89_{+1.17\%}$ | 1.65 | 0.25 | 8.61 |

Our adaptive method achieves substantial computational savings (42-56% hypothesis reduction) with minimal performance impact (+0.67-1.17% relative WER increase) in zero-shot settings.

*Experimental notes:* For CommonVoice, we evaluated hypothesis sets in the range $[1, 4]$ as preliminary tests showed performance degradation beyond 4 hypotheses; the GER baseline (11.73%) represents the best fixed size in this range. TedLium-3 was excluded from this evaluation because, consistent with findings in (Ma et al., 2024), all zero-shot working points performed worse than the baseline across all hypothesis sizes - likely due to the dataset's initially strong baseline performance, making the comparison uninformative.

These experiments validate three key findings: (1) the adaptive framework generalizes across model scales with consistent behavior, (2) zero-shot prompting achieves substantial efficiency gains (42-56% reduction) despite smaller absolute WER differences, and (3) computational savings persist relative to the increased inference cost of larger models.

### B.4 EXTENSION TO SPEECH TRANSLATION TASKS

To demonstrate the broader applicability of our adaptive hypothesis selection framework, we evaluated its performance on speech translation tasks using the GenTranslate paradigm Hu et al. (2024b). This cross-domain validation examines whether our method maintains its computational efficiency benefits when applied to translation scenarios involving N-best hypothesis integration.

**Dataset Selection and Monotonicity Validation.** We selected three language pairs from the FLEURS X→En speech translation dataset (fr→en, cy→en, ar→en) based on a critical prerequisite: monotonic performance improvement with increasing hypothesis set sizes. Using the published GenTranslate checkpoint [3], we validated that BLEU scores followed the expected ordering $\text{BLEU}(N = 5) > \text{BLEU}(N = 4) > ... > \text{BLEU}(N = 1)$ on average across these language pairs. This monotonicity condition ensures a meaningful performance-compute trade-off exists, validating the potential utility of adaptive selection.

**Methodological Adaptation for Speech Translation.** A key challenge emerged in adapting our framework to speech translation tasks: BLEU, the standard evaluation metric, operates at the corpus level and provides limited meaningful information at the instance level required for our risk-based selection mechanism. To address this, we employed TER (Translation Edit Rate) for instance-level risk computation and adaptive set selection, while reporting final results using corpus-level BLEU for comparability with existing work. This approach leverages TER's established validity at the instance level while maintaining evaluation consistency. We validated that TER and BLEU preserve relative ordering (with inverse correlation) across our test sets.

Table B.5: Speech translation results with adaptive hypothesis selection on FLEURS X→En test sets compared to fixed N=5 baseline. Our method achieves substantial computational savings while maintaining competitive translation quality.

| Task | GER | | Adaptive Selection | | | $\alpha(\%)$ | Success Rate | $\text{TER}_{O_{llm}}$ |
|---|---|---|---|---|---|---|---|---|
| | TER (%) | BLEU | Avg. Size | BLEU | TER (%) | | | |
| fr→en | 4.62 | 37.50 | $3.21_{35.8\%}$ | $37.23_{-0.70\%}$ | 4.67 | 5 | 0.97 | 4.24 |
| cy→en | 5.1 | 33.89 | $2.62_{47.7\%}$ | $33.39_{-1.47\%}$ | 5.3 | 6.9 | 0.96 | 4.67 |
| ar→en | 5.21 | 34.47 | $1.72_{65.5\%}$ | $33.44_{-2.98\%}$ | 5.29 | 5.65 | 0.98 | 4.81 |

**Implementation Details.** Due to smaller test set sizes (400-1000 samples) compared to ASR experiments, we increased the error tolerance $\delta$ to 0.3 to prevent overly conservative bounds in the LTT framework. All other methodological components, including parameter selection and risk calibration procedures, remained consistent with our ASR implementation. The same LTT sequential testing approach was applied with TER-based loss functions for hypothesis set selection.

**Summary.** This cross-domain validation demonstrates that our adaptive hypothesis selection framework generalizes effectively beyond ASR to speech translation tasks, achieving substantial

---

[3]https://huggingface.co/PeacefulData/GenTranslate

computational savings (36-66% hypothesis reduction) while maintaining competitive translation quality. These results confirm the broader applicability of our approach across generative error correction scenarios involving N-best hypothesis integration, addressing the computational efficiency challenges inherent in LLM-based post-processing systems.

## C  LLM TRAINING CONFIGURATION DETAILS

### C.1  HYPERPARAMETERS

We train using AdamW optimizer, effective batch size 32 (achieved through batch size 8 with 4-step gradient accumulation), and cosine learning rate scheduler (with 0.05 warmup ratio). The LoRA configuration uses rank $r = 16$ and scaling parameter $\alpha = 32$, implemented via the PEFT library (Mangrulkar et al., 2022).

Dataset-specific hyperparameters accommodate varying dataset sizes: learning rate range from 5e-5 to 1e-4, dropout rates range from 0.05-0.1, training epochs from 5-10, with larger datasets requiring higher values for those parameters to achieve optimal convergence.

### C.2  PROMPT TEMPLATE

The training utilizes the following prompt template:

> *"Correct this speech recognition transcript using the hypotheses below. Provide ONLY the corrected transcript, nothing more.*
> *###Hypotheses:*
> *- {1st ∼ 5th utterances}*
> *###Corrected-transcript:"*

### C.3  COMPUTATIONAL REQUIREMENTS

Model training is conducted on a single NVIDIA RTX 6000 Ada GPU with 48GB memory. Training duration varies by dataset size: CHiME-4 requires approximately 1 hour due to its smaller scale (9,600 samples), while TedLium-3 and CommonVoice each require 3-4 hours given their larger training sets (35,000 samples each). The LoRA parameterization significantly reduces computational overhead compared to full fine-tuning, enabling efficient adaptation while maintaining the frozen backbone parameters.

## D  RISK CONTROL FRAMEWORKS AND EXTENSIONS

### D.1  CONFORMAL PREDICTION FRAMEWORK

Conformal prediction is a distribution-free statistical framework that provides uncertainty quantification for machine learning predictions with finite-sample guarantees. Given a calibration dataset separate from training data, CP constructs prediction sets that satisfy coverage properties regardless of the underlying model architecture or data distribution.

Let $\mathcal{X}$ denote the input space and $\mathcal{Y}$ the output space. Consider a calibration set $\{(X^{(i)}, Y^{(i)})\}_{i=1}^{m}$ where $(X^{(i)}, Y^{(i)}) \in \mathcal{X} \times \mathcal{Y}$, and a new test point $(X^{(m+1)}, Y^{(m+1)})$. Conformal prediction requires that the calibration data and test point are exchangeable, meaning the joint distribution remains invariant under permutations.

For the test input $X^{(m+1)}$ with unknown label $Y^{(m+1)}$, the goal is to construct a prediction set $C(X^{(m+1)})$ such that:

$$P(Y^{(m+1)} \notin C(X^{(m+1)})) \leq \alpha, \tag{14}$$

where $\alpha$ is a user-specified significance level (e.g., 0.1 for 90% coverage).

The framework relies on nonconformity scores $s_i : \mathcal{X} \times \mathcal{Y} \to \mathbb{R}$ that measure how atypical a prediction is for a given input. For any pair $(x, y) \in \mathcal{X} \times \mathcal{Y}$, the nonconformity score $s(x, y)$ should reflect their agreement, with lower scores indicating better agreement. For the calibration set, where true labels are known, $s_i = s(X^{(i)}, Y^{(i)})$ quantifies the disagreement between the model's prediction and the true label. The prediction set is then constructed by including all labels whose nonconformity scores fall below a data-dependent threshold, computed based on the $(1-\alpha)$ quantile of the calibration scores.

### D.2 CONFORMAL RISK CONTROL APPROACH

#### D.2.1 BACKGROUND

Consider a calibration dataset $\{(X^{(i)}, Y^{(i)})\}_{i=1}^m$, where $X \in \mathcal{X}$ and $Y \in \mathcal{Y}$ denote feature-response pairs. Conformal risk control (CRC) extends CP to control the expectation of any bounded, monotone loss function $\ell : 2^{\mathcal{Y}} \times \mathcal{Y} \to [0, B]$:

$$E[\ell(\Gamma_\lambda(X^{(m+1)}), Y^{(m+1)})] \leq \alpha, \tag{15}$$

under the same notations represented in 5.

The key insight is that for monotone loss functions—where enlarging the prediction set cannot increase the loss—CRC maintains the distribution-free guarantees of standard CP while enabling control over task-specific risk measures. The CRC threshold selection procedure aims to find the optimal threshold:

$$\hat{\lambda} = \inf \left\{ \lambda : \frac{m}{m+1} \hat{R}_m(\lambda) + \frac{B}{m+1} \leq \alpha \right\}, \tag{16}$$

where the empirical risk is computed as:

$$\hat{R}_m(\lambda) = \frac{1}{m} \sum_{i=1}^m \ell(\Gamma_\lambda(X^{(i)}), Y^{(i)}), \tag{17}$$

representing the average empirical loss over the calibration set. For monotone loss functions, this threshold can be found efficiently by gradually adjusting $\lambda$ until the risk constraint is satisfied.

CRC provides finite-sample guarantees that are tight up to $O(1/m)$ terms as stated in the following Theorem.

**Theorem 2** (CRC Finite-Sample Guarantee). *Under the exchangeability assumption and for bounded monotone loss functions, the set predictor $\Gamma_{\hat{\lambda}}$ selected by the CRC procedure satisfies:*

$$\alpha - \frac{2B}{m+1} \leq E[\ell(\Gamma_{\hat{\lambda}}(X^{(m+1)}), Y^{(m+1)})] \leq \alpha. \tag{18}$$

Note that CRC reduces to standard CP when the loss function is the miscoverage indicator. CRC has been applied to areas such as medical diagnosis, autonomous driving, ordinal classification, and ranked retrieval systems (Andéol et al., 2023; Xu et al., 2023; 2024; Overman et al., 2024).

#### D.2.2 CRC IMPLEMENTATION AND EMPIRICAL ANALYSIS

We initially explored conformal risk control (CRC) as our primary theoretical framework for adaptive hypothesis selection. Our CRC implementation follows Algorithm D.1, using the relative WER degradation loss function defined in Equation 8 with calibrated thresholds to control expected performance degradation.

Table D.1 presents our CRC experimental results across the three datasets. The method achieves substantial computational savings while maintaining competitive performance. These results validate the practical effectiveness of adaptive selection based on this method.

---

**Algorithm D.1** Adaptive Selection Procedure with CRC Calibration Framework

---

**Require:** Calibration set $\{(\mathcal{H}_N^{(i)}, y^{(i)})\}_{i=1}^m$ and a test sample $\mathcal{H}_N^{(m+1)}$
1: **Calibration Phase:**
2: **for** $\lambda \in \Lambda$ (candidate threshold values) **do**
3:     Compute $\ell(\Gamma_\lambda(\mathcal{H}_N^{(i)}), y^{(i)})$ for all $i \in [m]$
4:     Estimate $\hat{R}_m(\lambda) = \frac{1}{m} \sum_{i=1}^m \ell(\Gamma_\lambda(\mathcal{H}_N^{(i)}), y_i)$
5: **end for**
6: Select $\hat{\lambda} = \inf \left\{ \lambda : \frac{m}{m+1} \hat{R}_m(\lambda) + \frac{B}{m+1} \leq \alpha \right\}$
7: **Test Phase:**
8: Compute normalized scores $\mathbf{s} = \mathrm{softmax}(\phi_\gamma(\mathbf{c}^{(m+1)})/\tau)$
9: Select $n^* = \min\{n : \sum_{i=1}^n s_i \geq \hat{\lambda}\}$
10: **Return** hypothesis set $\mathcal{H}_{n^*}^{(m+1)} = \left\{ \left( \hat{y}_1^{(m+1)}, c_1^{(m+1)} \right), \ldots, \left( \hat{y}_{n^*}^{(m+1)}, c_{n^*}^{(m+1)} \right) \right\}$

---

However, our analysis revealed that strict monotonicity is violated in approximately 20% of samples, where smaller hypothesis sets occasionally outperform larger ones. While 95% of consecutive pairwise comparisons maintain monotonicity, indicating predominantly monotonic behavior, these violations present a theoretical challenge for CRC's formal guarantees. We evaluated the monotonizing procedure proposed by Angelopoulos et al. (2024b), which constructs $\tilde{\ell}_i(\lambda) = \sup_{\lambda' \geq \lambda} \ell_i(\lambda')$ to enforce monotonicity. However, empirical results showed degraded performance across datasets, as monotonizing eliminates precisely the beneficial cases where smaller sets genuinely outperform larger ones—the phenomenon enabling our computational savings.

These findings indicate that monotonicity violations in ASR often signal exploitable efficiency opportunities rather than problematic cases. While our CRC implementation demonstrates strong empirical performance and effective risk control in practice, the theoretical violations prevent us from claiming formal statistical guarantees. Therefore, we present this CRC approach as an additional empirical demonstration that complements our theoretically rigorous LTT framework, which naturally handles non-monotone losses while providing formal $P(R(\hat{\lambda}) \leq \alpha) \geq 1 - \delta$ guarantees.

Table D.1: WER (%) results with LLaMA-2-7B fine-tuning. Baseline: Whisper's top-1 hypothesis. $\mathrm{O}_{llm}$: post-LLM oracle. Our method results represent one operating point from Figure 2. Subscript percentages denote relative WER change vs. vanilla GER (WER column) and relative size reduction vs. constant $N = 5$ (size column).

| Test Set | Baseline | GER | Our Method | | $\alpha(\%)$ | Target WER | $\mathrm{O}_{llm}$ |
|---|---|---|---|---|---|---|---|
| | | | Set Size | WER | | | |
| TedLium-3 | 8.0 | 6.04 | $2.145_{57.1\%}$ | $5.96_{-1.3\%}$ | 1.785 | 6.115 | 4.33 |
| CHiME-4 | 11.49 | 6.38 | $3.8_{24.0\%}$ | $6.55_{+2.7\%}$ | 1.9 | 6.63 | 4.73 |
| CommonVoice | 14.1 | 8.46 | $3.1_{38.0\%}$ | $8.5_{+0.5\%}$ | 1.7 | 8.65 | 6.95 |

## D.3 MULTI-PARAMETER OPTIMIZATION VIA PARETO TESTING

### D.3.1 MOTIVATION AND METHODOLOGY

Our main LTT framework fixes normalization parameter $\gamma$ and temperature $\tau$ based on dataset characteristics, then optimizes threshold $\lambda$ online. While effective, this approach requires pre-deployment validation analysis to determine appropriate parameter values. We explore an extension based on Pareto Testing (Laufer-Goldshtein et al., 2023) that jointly optimizes all three parameters $(\gamma, \tau, \lambda)$ simultaneously, eliminating the need for dataset-specific parameter selection while potentially discovering superior performance-efficiency trade-offs through expanded degrees of freedom.

The Pareto Testing framework operates in two stages: first, multi-objective optimization on calibration subset *calib1* constructs a Pareto frontier of parameter combinations that optimize both relative WER degradation and computational cost; second, Fixed Sequential Testing on *calib2* applies statistical validation to ensure risk control guarantees. This maintains theoretical rigor while enabling automated deployment across diverse acoustic conditions.

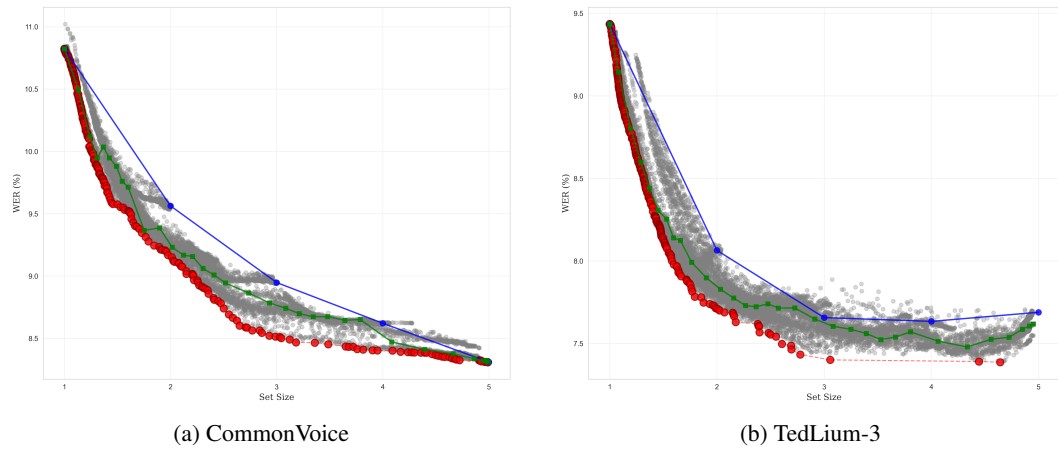

(a) CommonVoice                                    (b) TedLium-3

Figure D.1: Pareto frontier visualization for CommonVoice and TedLium-3. Gray points represent all feasible working points in the $(\gamma, \tau, \lambda)$ parameter space. Green points indicate valid LTT working points across different $\lambda$ values, and the red curve depicts the extracted Pareto frontier from a representative calibration split, showing non-dominated combinations in WER vs. set size space.

### D.3.2 EXPERIMENTAL RESULTS

We evaluated this approach on TedLium-3 and CommonVoice datasets. To extract the Pareto frontier, we split the calibration set into two subsets: the first is used for multi-objective optimization to construct the frontier of non-dominated $(\gamma, \tau, \lambda)$ combinations, and the second for statistical validation via sequential testing.

The expanded parameter space enables configurations that our heuristic selection cannot identify, particularly intermediate settings that effectively balance both objectives, as evidenced by the gap between the green and red curves in Figure D.1 - where the Pareto frontier (red) consistently achieves better WER-efficiency trade-offs than the individual LTT working points (green).

We then compare the Pareto-selected working points against all fixed set size baselines ($N = 1$ through $N = 5$) across varying $\alpha$ levels. As shown in Figure D.2, our joint optimization achieves superior performance-efficiency trade-offs across all operating points, matching or exceeding the results of our main LTT approach (Table D.2), confirming that the adaptive selection mechanism is robust across both optimization strategies.

Table D.2: Pareto WER (%) results. Baseline: Whisper's top-1 hypothesis. $O_{llm}$: post-LLM oracle. Results are averaged over $T = 30$ independent trials. Subscript percentages denote relative WER change vs. vanilla GER (WER column) and relative size reduction vs. constant $N = 5$ (size column).

| Test Set | Baseline | GER | LTT Method | | $\alpha(\%)$ | $\delta$ | Success Rate | $O_{llm}$ |
|---|---|---|---|---|---|---|---|---|
| | | | Set Size | WER | | | | |
| TedLium-3 | 9.3 | 7.53 | $2.42_{-51.6\%}$ | $7.49_{-0.53\%}$ | 2.4 | 0.10 | 0.92 | 5.58 |
| CommonVoice | 12.44 | 8.32 | $3.18_{-36.4\%}$ | $8.54_{+2.64\%}$ | 1.95 | 0.10 | 0.93 | 6.96 |

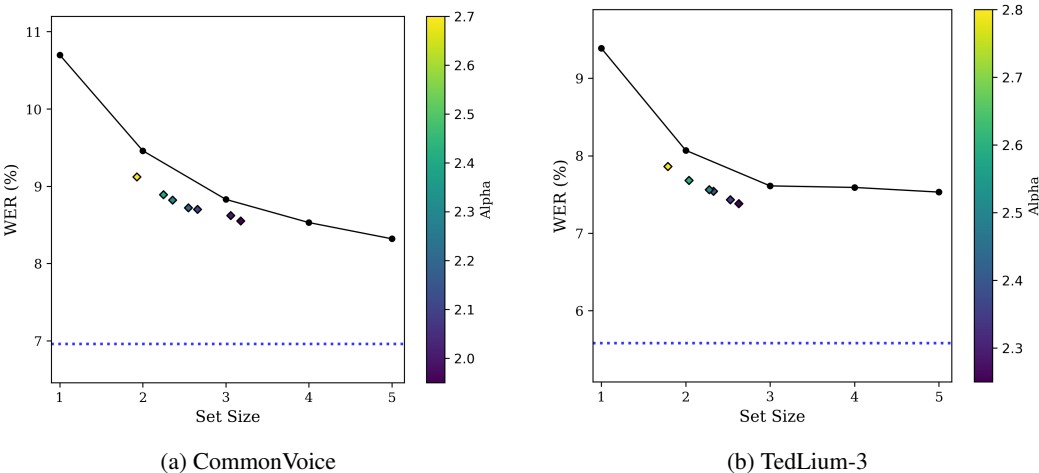

(a) CommonVoice  (b) TedLium-3

Figure D.2: Performance-compute trade-off comparison for CommonVoice and TedLium-3. Each plot shows WER vs. set size for fixed set size baselines, alongside the Pareto-selected working points across varying $\alpha$ levels, demonstrating superior trade-offs achieved through joint parameter optimization.

### D.3.3  DISCUSSION AND CONCLUSION

This extension transforms our framework from requiring dataset-specific analysis to fully automated parameter optimization. The primary trade-off is computational overhead during calibration, as the two-stage optimization evaluates multiple parameter combinations before statistical testing. However, this occurs only during setup, with identical inference costs thereafter.

Pareto Testing represents a step toward adaptive ASR systems that automatically configure themselves across diverse environments without domain expertise. While our main approach remains effective for characterized conditions, this extension provides deployment flexibility and potential performance gains through joint optimization of all hyperparameters while maintaining identical theoretical guarantees.

