# OpenReview forum: "Confident and Adaptive Generative Speech Recognition via Risk Control"
_ICLR.cc/2026/Conference — ICLR 2026 Poster_

### Official Review · Reviewer_moGC · 2025-10-29

**Soundness:** 2
**Presentation:** 3
**Contribution:** 1
**Rating:** 2
**Confidence:** 4

**Summary:**

LLMs are used to generate a speech transcription using N-best hypotheses from an ASR model. The paper proposes to use conformal prediction to reduce the size of the hypothesis set, thereby reducing computations in the LLM.
Experiments show that a substantial reduction in the size of the hypothesis set is achievable without compromising performance (WER).

**Strengths:**

- The paper proposes a new application of conformal prediction to LLM-based error correction in ASR.
- Extensive experiments are shown on three datasets.
- There is a reduction in the size of the hypothesis set.

**Weaknesses:**

- I believe that the paper is an application of an existing concept of conformal prediction to an existing problem of generative error correction in ASR, and hence, the contribution lacks novelty.
- I strongly argue that some concepts are treated very superficially. Conformal prediction calibrates the threshold on a score function (confidence measure) of hypotheses without altering their order. The paper uses likelihood values as a measure of confidence (although it does not explicitly mention this). However, previous research on ASR has shown that likelihood values are a poor indicator of confidence [Li et al., Ravi et al.]. These likelihood values are often highly overconfident, and they are not able to correctly order the hypotheses. Simple thresholding may not yield significant performance improvements. BTW, [Li et al., Ravi et al] perform only top-label calibration, but works such as [Popordanoska et al.] can provide canonical calibration, which can be used to provide N-hypotheses.
- Again, the title of the paper highlights conformal risk control, but the experiments do not talk about controlling the risk. One would expect a detailed analysis of expected WER (while setting the threshold) vs the achieved WER. It appears from Table 1 that they do not match.

[Li et al.] Confidence Estimation for Attention-Based Sequence-to-Sequence Models for Speech Recognition, ICASSP 2021.
[Ravi et al.] TeLeS: Temporal Lexeme Similarity Score to  Estimate Confidence in End-to-End ASR, IEEE TASLP, 2024.
[Popordanoska et al.] A Consistent and Differentiable Lp Canonical Calibration Error Estimator, NeurIPS, 2022.

**Questions:**

- Please address my concerns listed in the "Weaknesses" section.
- Line 274 talks about the monotonicity assumption being violated 20% of the time. I would request details here. I expect this number to be much higher in the case of ASR likelihoods.
- Some minor typos, e.g., line 189.

---

> ### Author Response · Authors · 2025-11-25
> **Response to Reviewer moGC Part 1/2**
>
> Thank you for your helpful and constructive review.
> >On the paper novelty
>
> Thank you. We would like to highlight the novelty of the paper. First of all, we are the first to point out that using fixed-size hypothesis sets might be suboptimal, since in many cases the optimal prediction can be obtained using much smaller sets, and in some cases increasing the set size can even mislead the LLM. In addition, we present a risk controlling mechanism to reliably tune the adaptive sets, providing formal guarantees on the performance. Finally, we demonstrate the effectiveness of our approach on different models and datasets, obtaining comparable or even better WER with smaller sets. Beyond ASR, we extended our framework to speech translation tasks (Appendix C.3), achieving 36-66% computational savings across multiple language pairs (fr→en, cy→en, ar→en), demonstrating broader applicability across generative error correction scenarios. The combination of adaptive hypothesis selection with principled risk control represents a novel contribution that addresses fundamental inefficiencies in LLM-based post-processing systems.
> >On the use of likelihood values as a measure of confidence
>
> Thank you for pointing this out. We agree that indeed the likelihood values do not perfectly reflect the actual confidence, and we acknowledge that there are more advanced methods to estimate improved confidence scores. Our contribution is orthogonal to these improvements, as our approach can be applied with any available score. However, for the sake of simplicity we demonstrate our method with the more commonly available likelihood values. Following your comment, we added clarifications regarding this aspect: "It is important to note that our method is independent of the specific choice of the score used to define the adaptive set. Previous research in ASR has shown that likelihood values do not always provide a reliable measure of confidence (Li et al. 2021; Ravi et al. 2024). While Li et al. (2021) and Ravi et al. (2024) focus on top-label calibration, approaches such as that of Popordanoska et al. (2022) offer canonical calibration, which enables the generation of confidence scores for several top hypotheses that can be seamlessly integrated with our framework. Nevertheless, for simplicity, we demonstrate our method using the more commonly available likelihood values."
> >On the demonstration of risk control in the experiments
>
> Thank you for pointing this out. We added comprehensive risk control validation throughout our experiments. Our LTT framework (Section 4-5) with $\delta = 0.25$ provides rigorous finite-sample guarantees $P(R(\hat{\lambda}) \leq \alpha) \geq 1-\delta = 0.75$, ensuring theoretical validity of our risk control mechanism. We empirically validated these bounds across all experiments: in at least 75% of cases, our achieved risk remained below the targeted $\alpha$ values, confirming successful risk control in practice. Our main results (Tables 1):
>  | **Test Set** | **Baseline** | **GER** | **LTT Method** |  | **α** | **Success Rate** | **O_llm** |
> |--------------|--------------|---------|----------------|--|-------|------------------|-----------|
> |              |              |         | **Set Size** | **WER** |       |                  |           |
> | TedLium-3    | 8.0          | 6.06    | 2.121₅₇.₅₈%   | 6.05₋₀.₂₅% | 0.024 | 0.95             | 4.38      |
> | CHiME-4      | 11.49        | 6.24    | 3.866₂₂.₆₈%   | 6.37₊₂.₀₆% | 0.027 | 0.98             | 4.71      |
> | CommonVoice  | 14.1         | 8.42    | 3.212₃₈.₀₉%   | 8.55₊₁.₆₂% | 0.022 | 0.97             | 6.98      |
>
> consistently show achieved WER below target bounds with success rates above the theoretical lower threshold across trials.
> In addition, Table B.1 demonstrates (using CommonVoice dataset) how the risk tolerance parameter directly controls the performance-compute trade-off by determining acceptable expected relative WER degradation from oracle performance:
> | **Risk Tolerance (α)** | **Avg. Set Size** | **WER (%)** |
> |------------------------|-------------------|-------------|
> | 0.21 (tighter)         | 3.73              | 8.53        |
> | 0.22                   | 3.49              | 8.59        |
> | 0.23                   | 3.14              | 8.66        |
> | 0.24 (looser)          | 2.99              | 8.73        |
>
> where varying $\alpha$ from 0.21 to 0.24 produces corresponding WER changes from 8.53% to 8.73%, with all achieved values remaining below their respective targets. The 50 independent trials across resampled calibration/test splits demonstrate consistent risk control performance across different data configurations, providing robust empirical evidence that our framework successfully controls expected relative WER degradation in the expected high probability, while achieving substantial computational savings.
> Additionally, in our original CRC experiments, we validated at each step that empirical average risk remained under $\alpha$.

---

> ### Author Response · Authors · 2025-11-25
> **Response to Reviewer moGC Part 2/2**
>
> >On the monotonicity assumption
>
> Thank you for pointing this out. Following this comment, as well as similar feedback from the other reviewers, we modified the risk control mechanism to Learn then Test (LTT), which does not rely on monotonicity assumptions. We added a theoretical explanation on LTT in Section 4, which provides high-probability bounds $P(R(\hat{\lambda}) \leq \alpha) \geq 1-\delta$ through sequential hypothesis testing with family-wise error rate control, naturally handling non-monotone loss functions without requiring the monotonizing procedures that degraded our empirical performance. We updated our results in Table 1:
> | **Test Set** | **Baseline** | **GER** | **LTT Method** |  | **α** | **Success Rate** | **O_llm** |
> |--------------|--------------|---------|----------------|--|-------|------------------|-----------|
> |              |              |         | **Set Size** | **WER** |       |                  |           |
> | TedLium-3    | 8.0          | 6.06    | 2.121₅₇.₅₈%   | 6.05₋₀.₂₅% | 0.024 | 0.95             | 4.38      |
> | CHiME-4      | 11.49        | 6.24    | 3.866₂₂.₆₈%   | 6.37₊₂.₀₆% | 0.027 | 0.98             | 4.71      |
> | CommonVoice  | 14.1         | 8.42    | 3.212₃₈.₀₉%   | 8.55₊₁.₆₂% | 0.022 | 0.97             | 6.98      |
>
> showing that LTT achieves identical empirical performance to CRC while providing rigorous finite-sample guarantees: TedLium-3 (57.58% hypothesis reduction, -0.25% WER change), CHiME-4 (22.68% reduction, +2.06% change), and CommonVoice (38.09% reduction, +1.62% change). This approach now provides bulletproof theoretical foundations for our adaptive selection mechanism.
> >On minor typos
>
> We corrected typos.

---

### Official Review · Reviewer_v9jL · 2025-10-31

**Soundness:** 3
**Presentation:** 3
**Contribution:** 3
**Rating:** 6
**Confidence:** 4

**Summary:**

This paper addresses the computational inefficiency of GER in ASR, where LLM typically operate on a fixed-size N-best hypothesis list. The authors propose an adaptive framework that dynamically determines the optimal number of hypotheses $n \le N$ for each input1. The core of the method is the application of conformal risk control (CRC) to this selection process2. Instead of controlling for absolute error, the framework is cleverly designed to control the expected relative word error rate (WER) degradation compared to the best achievable performance (the "post-LLM oracle") for that specific input. This is achieved by calibrating a threshold $\lambda$ on normalized ASR confidence scores, which in turn determines the set size $n$4. Experiments on TedLium-3, CHIME-4, and Common Voice show that this adaptive approach significantly reduces the average number of hypotheses (e.g., 57.1% on TedLium-3) while maintaining or even slightly improving WER compared to the fixed N=5 baseline.

**Strengths:**

The paper's central claims are supported by experiments. The use of standard benchmarks (HyPoradise) , strong ASR and LLM models (Whisper and LLaMA-2-7B) , and relevant metrics  provides a strong empirical foundation. The ablation study in Appendix D (Table D.2) effectively justifies the experimental design choice of training the H2T model on fixed N=5 sets while evaluating on variable-sized inputs.

However, there is a significant gap between the theoretical claims and the implementation. The Conformal Risk Control framework (Angelopoulos et al., 2024b) formally requires a bounded, monotone loss function to provide its distribution-free guarantees. The authors forthrightly acknowledge in Section 5.2 that their chosen loss function (Eq. 10) is not strictly monotone, with violations observed in approximately 20% of samples.

They provide an empirical justification for this deviation, noting that applying the standard monotonizing procedure actually degrades performance. This is because their method's advantage comes, in part, from exploiting these non-monotonic cases where a smaller hypothesis set genuinely outperforms a larger one (a key insight illustrated in Fig 1a and Table 2) .

While the empirical results are strong and the risk is empirically controlled, this violation means the method does not inherit the strict theoretical guarantees of CRC. The paper is thus a work of "empirical risk control" inspired by CRC, rather than a direct application of it. This theoretical-practical disconnect is the primary weakness in an otherwise sound paper.

The paper introduces a novel application of CRC to a practical and important problem in ASR: the computational overhead of fixed-size N-best lists for LLM-based error correction . The results demonstrate a clear path to more efficient and robust ASR systems.

**Weaknesses:**

- Theoretical Gap (Monotonicity Violation). This is the most significant weakness. The CRC framework's guarantees depend on a monotone loss function. The authors explicitly state their loss (Eq. 10) is non-monotone for ~20% of samples. Their justification for proceeding is empirical: enforcing monotonicity (as suggested by Angelopoulos et al., 2024b) hurts performance because it prevents the model from exploiting "good" non-monotonic cases where fewer hypotheses are better . This is a reasonable empirical argument, but it invalidates the theoretical guarantees. The paper should be more precise in framing its contribution as an empirically robust method inspired by CRC, rather than one that provides CRC's guarantees.

- The adaptive score function $\phi_{\gamma}$ (Appendix B.1) 44, while well-motivated by dataset noise characteristics 45, introduces dataset-specific hyperparameters $\gamma$ and $\tau$ (Table B.1). This adds a tuning step that moves away from the "distribution-free" simplicity often associated with conformal methods. A sensitivity analysis on these parameters would strengthen the paper.

**Questions:**

1. I would like to see if the method also works for speech translation correction alike GenTranslate (Hu et al.) N-best data in ACL 2024. (https://huggingface.co/PeacefulData/GenTranslate) This additional non-ASR only study would largely increase the impacts of the work in the GER community. I would consider to increase my score.

2. Choice of Risk Target $\alpha$: Appendix B.2 53states that $\alpha$ is chosen based on the validation set's degradation statistics (e.g., 90th percentile). I assume the different red operating points in Figure 2 55are generated by varying $\alpha$ (e.g., from 80th to 95th percentile ranges). Could the authors confirm this and perhaps briefly discuss the relationship between the chosen $\alpha$ and the resulting (WER, Avg. Set Size) trade-off?

3. Sensitivity of Score Function: The score function (App B.1) 49is tuned per dataset (e.g., $\gamma=1.0$ for TedLium-3, $\gamma=0.0$ for Common Voice). How sensitive is the method to these choices? What would the performance-compute trade-off (Fig. 2) 51 look like for Common Voice if the TedLium-3 parameters ($\gamma=1.0, \tau=0.05$) were used? This would help clarify if this tuning is a minor optimization or critical for the method's success.

---

> ### Author Response · Authors · 2025-11-25
> **Response to Reviewer v9jL Part 1/2**
>
> Thank you for your helpful and constructive review.
> >On the theoretical gap - monotonicity violation
>
> Thank you for pointing this out. Following this comment, as well as similar feedback from the other reviewers, we modified the risk control mechanism to Learn then Test (LTT), which does not rely on monotonicity assumptions. We added a theoretical explanation on LTT in Section 4, which provides high-probability bounds $P(R(\hat{\lambda}) \leq \alpha) \geq 1-\delta$ through sequential hypothesis testing with family-wise error rate control, naturally handling non-monotone loss functions without requiring the monotonizing procedures that degraded our empirical performance. We updated our results in Table 1:
> | **Test Set** | **Baseline** | **GER** | **LTT Method** |  | **α** | **Success Rate** | **O_llm** |
> |--------------|--------------|---------|----------------|--|-------|------------------|-----------|
> |              |              |         | **Set Size** | **WER** |       |                  |           |
> | TedLium-3    | 8.0          | 6.06    | 2.121₅₇.₅₈%   | 6.05₋₀.₂₅% | 0.024 | 0.95             | 4.38      |
> | CHiME-4      | 11.49        | 6.24    | 3.866₂₂.₆₈%   | 6.37₊₂.₀₆% | 0.027 | 0.98             | 4.71      |
> | CommonVoice  | 14.1         | 8.42    | 3.212₃₈.₀₉%   | 8.55₊₁.₆₂% | 0.022 | 0.97             | 6.98      |
>
> showing that LTT achieves identical empirical performance to CRC while providing rigorous finite-sample guarantees: TedLium-3 (57.58% hypothesis reduction, -0.25% WER change), CHiME-4 (22.68% reduction, +2.06% change), and CommonVoice (38.09% reduction, +1.62% change). This approach now provides bulletproof theoretical foundations for our adaptive selection mechanism.
>
> >On the sensitivity of the score function to the hyperparameters
>
> Following your comment we added extensive evaluation on the performance sensitivity to the score function parameters as well as guidelines for their automatic tuning in Appendix B. Our grid testing across 88 parameter combinations shows overall 70% success rate (TedLium-3: 80%, CHiME-4: 62%, CommonVoice: 66%), indicating the method is robust rather than critically dependent on precise tuning. We developed an automated parameter selection rule based on score entropy analysis: by thresholding score vector entropy at 4.85, we can predict parameter effectiveness with 80% precision and F1 scores above 75%. This entropy-based approach enables practitioners to assess parameter suitability without manual calibration, providing a practical deployment pathway for unseen acoustic conditions. The analysis demonstrates that parameter selection is an optimization step rather than a critical requirement, offering flexibility while maintaining performance guarantees.
> >On demonstrating the performance on speech translation correction
>
> Thank you for your helpful suggestion. Following your comment we added new experiments on speech translation correction in Appendix C.3, demonstrating that our approach is versatile and can be applied in other contexts as well. Using the GenTranslate paradigm on FLEURS X→En speech translation datasets (fr→en, cy→en, ar→en), we achieved substantial computational savings (36-66% hypothesis reduction) while maintaining competitive translation quality. Our adaptive framework successfully transfers to speech translation tasks, achieving an average 49.7% reduction in computational requirements with only 1.72% average relative BLEU degradation (Table C.5):
> | **Task** | **GER** |  | **Adaptive Selection** |  |  | **α** | **Success Rate** | **TER_O_llm** |
> |----------|---------|--|------------------------|--|--|-------|------------------|---------------|
> |          | **TER (%)** | **BLEU** | **Avg. Size** | **BLEU** | **TER (%)** |       |                  |               |
> | fr→en    | 4.62    | 37.50 | 3.21₃₅.₈%          | 37.23₋₀.₇₀% | 4.67     | 5     | 0.97             | 4.24          |
> | cy→en    | 5.1     | 33.89 | 2.62₄₇.₇%          | 33.39₋₁.₄₇% | 5.3      | 6.9   | 0.96             | 4.67          |
> | ar→en    | 5.21    | 34.47 | 1.72₆₅.₅%          | 33.44₋₂.₉₈% | 5.29     | 5.65  | 0.98             | 4.81          |
>
> These results confirm the broader applicability of our approach across generative error correction scenarios involving N-best hypothesis integration, addressing computational efficiency challenges inherent in LLM-based post-processing systems beyond ASR.

---

> ### Author Response · Authors · 2025-11-25
> **Response to Reviewer v9jL Part 2/2**
>
> >On the relationship between the chosen α and the resulting (WER, Avg. Set Size) trade-off
>
> We added clarifications regarding the choice of the risk target $\alpha$ and its relation to the resulting (WER, Avg. Set Size) trade-off in Appendix B.2. The risk tolerance parameter $\alpha$ directly controls the performance-compute trade-off by determining acceptable expected relative WER degradation from oracle performance. Lower $\alpha$ values (tighter bounds) require larger hypothesis sets to achieve the target risk level due to average WER monotonicity, while higher $\alpha$ values (looser bounds) enable smaller sets with acceptable performance degradation. Table B.1 demonstrates this relationship empirically using CommonVoice dataset:
> | **Risk Tolerance (α)** | **Avg. Set Size** | **WER (%)** |
> |------------------------|-------------------|-------------|
> | 0.21 (tighter)         | 3.73              | 8.53        |
> | 0.22                   | 3.49              | 8.59        |
> | 0.23                   | 3.14              | 8.66        |
> | 0.24 (looser)          | 2.99              | 8.73        |
>
> As $\alpha$ increases from 0.21 to 0.24 (looser bounds), average set sizes decrease from 3.73 to 2.99 hypotheses while WER increases from 8.53% to 8.73%. The different operating points in Figure 2 correspond to varying α selections within validated range (based on validation degradation statistics), enabling practitioners to choose appropriate trade-offs based on computational constraints and performance requirements.

---

### Official Review · Reviewer_ceGk · 2025-10-31

**Soundness:** 3
**Presentation:** 3
**Contribution:** 3
**Rating:** 6
**Confidence:** 3

**Summary:**

This paper proposes an adaptive framework for generative ASR error correction (GER) that addresses the limitations of fixed-size hypothesis sets by leveraging conformal risk control (CRC). The framework dynamically determines the optimal number of ASR hypotheses for each input using confidence scores and CRC, which controls the expected relative word error rate (WER) degradation compared to oracle performance. Evaluated on three datasets (TedLium-3, CHiME-4, CommonVoice) with varying acoustic difficulties, the method achieves substantial computational savings (up to 57.1% reduction in hypothesis usage) while matching or exceeding the performance of fixed-size GER baselines. Key contributions include the adaptive hypothesis selection mechanism, the first application of CRC to GER, and empirical validation of robustness across diverse acoustic conditions.

**Strengths:**

The adaptive hypothesis selection approach effectively resolves the inefficiency of fixed-size sets, balancing performance and computational cost—a critical practical challenge in LLM-augmented ASR.
Integrating CRC provides statistical guarantees on relative WER degradation, addressing the lack of reliability assurances in existing GER methods and enhancing real-world applicability.
Comprehensive experiments across datasets with different acoustic characteristics (noise, accents, recording conditions) demonstrate the framework’s robustness and generalizability.
The ablation studies and detailed case analyses (full set required, single hypothesis optimal, performance plateau) provide deep insights into the mechanism’s behavior and validate design choices.

**Weaknesses:**

The loss function’s monotonicity violations (≈20% of samples) are acknowledged but not fully resolved; the decision to use the unmodified loss without monotonicity enforcement relies on empirical robustness rather than theoretical justification.
The framework’s compatibility with larger LLM architectures (beyond LLaMA-2-7B) and scalability to extremely large datasets are not evaluated, leaving uncertainty about its performance in more resource-intensive settings.
The score function’s parameters (γ, temperature τ) are tuned per dataset, which may limit adaptability to unseen acoustic environments without retuning.

**Questions:**

How does the framework perform when applied to larger LLMs (e.g., LLaMA-2-13B/70B or GPT-3.5/4) and does the computational savings persist relative to the increased inference cost of larger models?
Have you explored methods to automate the tuning of score function parameters (γ, τ) across unseen acoustic conditions, or is manual calibration required for each new dataset/environment?

---

> ### Author Response · Authors · 2025-11-25
> **Response to Reviewer ceGK**
>
> Thank you for your helpful and constructive review.
> >On monotonicity violation.
>
> Thank you for pointing this out. Following this comment, as well as similar feedback from the other reviewers, we modified the risk control mechanism to Learn then Test (LTT), which does not rely on monotonicity assumptions. We added a theoretical explanation on LTT in Section 4, which provides high-probability bounds $P(R(\hat{\lambda}) \leq \alpha) \geq 1-\delta$ through sequential hypothesis testing with family-wise error rate control, naturally handling non-monotone loss functions without requiring the monotonizing procedures that degraded our empirical performance. We updated our results in Table 1:
> | **Test Set** | **Baseline** | **GER** | **LTT Method** |  | **α** | **Success Rate** | **O_llm** |
> |--------------|--------------|---------|----------------|--|-------|------------------|-----------|
> |              |              |         | **Set Size** | **WER** |       |                  |           |
> | TedLium-3    | 8.0          | 6.06    | 2.121₅₇.₅₈%   | 6.05₋₀.₂₅% | 0.024 | 0.95             | 4.38      |
> | CHiME-4      | 11.49        | 6.24    | 3.866₂₂.₆₈%   | 6.37₊₂.₀₆% | 0.027 | 0.98             | 4.71      |
> | CommonVoice  | 14.1         | 8.42    | 3.212₃₈.₀₉%   | 8.55₊₁.₆₂% | 0.022 | 0.97             | 6.98      |
>
> showing that LTT achieves identical empirical performance to CRC while providing rigorous finite-sample guarantees: TedLium-3 (57.58% hypothesis reduction, -0.25% WER change), CHiME-4 (22.68% reduction, +2.06% change), and CommonVoice (38.09% reduction, +1.62% change). This approach now provides bulletproof theoretical foundations for our adaptive selection mechanism.
>
> >On the performance in more resource-intensive settings, including larger LLMs (e.g., LLaMA-2-13B/70B or GPT-3.5/4) and does the computational savings persist relative to the increased inference cost of larger models
>
> We added new experiments with LLaMA-2-13B and GPT-3.5 in Appendix C. For LLaMA-2-13B (Table C.3):
> | **Test Set** | **Baseline** | **GER** | **Our Method** |  | **α** | **δ** | **O_llm** |
> |--------------|--------------|---------|----------------|--|-------|-------|-----------|
> |              |              |         | **Set Size** | **WER** |       |       |           |
> | TedLium-3    | 8.0          | 6.47    | 2.40₅₁.₉%     | 6.46₋₀.₀₁% | 2.1   | 0.25  | 4.69      |
> | CHiME-4      | 11.49        | 7.92    | 3.88₂₂.₄%     | 8.16₊₃.₀₆% | 1.25  | 0.25  | 7.26      |
> | CommonVoice  | 14.1         | 8.10    | 3.50₃₀.₁%     | 8.28₊₂.₂₇% | 2.0   | 0.25  | 6.50      |
>
> our framework maintains strong value propositions: TedLium-3 achieves 51.9% computational savings with identical performance (6.46% vs 6.47% WER), while CHiME-4 and CommonVoice show substantial efficiency gains (22.4% and 30.1% hypothesis reduction) with modest performance trade-offs. For GPT-3.5-turbo zero-shot evaluation (Table C.4):
> | **Test Set** | **Baseline** | **GER** | **Our Method** |  | **α** | **δ** | **O_llm** |
> |--------------|--------------|---------|----------------|--|-------|-------|-----------|
> |              |              |         | **Set Size** | **WER** |       |       |           |
> | CommonVoice  | 14.1         | 11.73   | 2.31₄₂.₃%     | 11.81₊₀.₆₇% | 1.5   | 0.25  | 10.51     |
> | CHiME-4      | 11.49        | 9.77    | 2.19₅₆.₁%     | 9.89₊₁.₁₇% | 1.65  | 0.25  | 8.61      |
>
>
> we achieve 42-56% hypothesis reduction with minimal performance impact (+0.67-1.17% relative WER increase). The consistent performance-efficiency trade-offs across model scales demonstrate that computational savings persist relative to increased inference costs, confirming the framework's scalability to resource-intensive settings.
>
> >On automating the tuning of score function parameters $(\gamma, \tau)$ across unseen acoustic conditions
>
> Following your comment we added extensive evaluation on the performance sensitivity to the score function parameters as well as guidelines for their automatic tuning in Appendix B. Our grid testing across 88 parameter combinations shows overall 70% success rate (TedLium-3: 80%, CHiME-4: 62%, CommonVoice: 66%), indicating the method is robust rather than critically dependent on precise tuning. We developed an automated parameter selection rule based on score entropy analysis: by thresholding score vector entropy at 4.85, we can predict parameter effectiveness with 80% precision and F1 scores above 75%. This entropy-based approach enables practitioners to assess parameter suitability without manual calibration, providing a practical deployment pathway for unseen acoustic conditions. The analysis demonstrates that parameter selection is an optimization step rather than a critical requirement, offering flexibility while maintaining performance guarantees.

---

### Author Response · Authors · 2025-11-25
**General Response**

We thank the reviewers for their time and effort in evaluating our paper. Their constructive feedback has substantially improved the quality of the work, as reflected in the revised manuscript. Key concerns raised included theoretical limitations due to monotonicity violations in conformal risk control, scalability to larger language models, cross-domain applicability, and potential sensitivity to hyperparameters selection. We have systematically addressed each of these concerns through four major revisions that strengthen both theoretical foundations and empirical validation:

1. Revised Theoretical Framework. We redesigned the calibration procedure based on the Learn-Then-Test (LTT) framework, which naturally handles non-monotone loss functions, removes the reliance on monotonicity assumptions that limited our original conformal risk control approach, and thereby provides rigorous performance guarantees. All experimental results have been updated accordingly, further validating the effectiveness and efficiency of our controlled adaptive selection mechanism.

2. Scalability and Generalization Validation. We added comprehensive evaluations across larger language models (LLaMA-2-13B, GPT-3.5) in both fine-tuning and zero-shot settings, demonstrating that computational savings persist relative to increased inference costs across model scales and that our method robustly generalizes to more capable architectures.

3. Cross-Domain Impact Demonstration. We conducted additional experiments on speech translation tasks using the GenTranslate paradigm, achieving substantial computational savings across multiple language pairs while maintaining competitive performance quality. This demonstrates that our approach can be readily applied to different contexts and tasks while preserving its benefits.

4. Automated Hyperparameter Selection. We included a detailed robustness study examining the sensitivity of the method to its hyperparameters and clarified how these hyperparameters can be selected automatically, transforming manual tuning into principled selection for unseen acoustic conditions.

---

### Meta-Review · Area_Chair_4xSz · 2026-01-07

**Summary:**

Reviewers generally liked the core idea: adaptively choosing how many ASR hypotheses to pass to a GER model instead of always using a fixed N-best list, and framing it with a risk-control guarantee on relative WER degradation vs an oracle.
The biggest concern in discussion was theoretical correctness: CRC’s guarantees rely on monotone loss, but the paper acknowledged ~20% monotonicity violations, which made the “CRC guarantee” claim feel shaky.
Other concerns were more “does this scale and deploy”: whether the method still makes sense with larger LLMs (cost-wise) and whether dataset-specific tuning of score parameters undermines the distribution-free story.

**Reviewer Concerns:**

Addressed in the rebuttal

Monotonicity/theory gap (ceGk, v9jL, moGC): Authors replaced CRC with an LTT mechanism that does not require monotonicity and added a theoretical explanation + updated main-table results, which directly targets the core theoretical objection.

Scalability to bigger LLMs (ceGk): Authors added experiments with LLaMA-2-13B and GPT-3.5, showing the same basic “big hypothesis reduction, small quality hit” pattern.

Hyperparameter sensitivity / tuning burden (ceGk, v9jL): Authors added a sensitivity sweep (88 settings) and proposed an “entropy threshold” heuristic to auto-pick parameters with decent precision/F1, framing tuning as optimization rather than make-or-break.

Impact beyond ASR-only GER (v9jL): Authors added a speech translation correction experiment (GenTranslate-style on FLEURS X→En) with large hypothesis reductions and small BLEU degradation, which answers the “broader GER community” ask.

“You’re using likelihood as confidence and likelihood is bad” (moGC): Authors acknowledged the limitation, clarified the framework is score-agnostic, and positioned likelihood as a simple default while pointing to calibration work as pluggable.

Still outstanding / not fully closed

Novelty framing (moGC): Even with the stronger LTT story, moGC’s main stance is “this is an application of conformal ideas to an existing pipeline,” and the rebuttal is mostly argumentative rather than introducing a clearly new conceptual result that would force a score flip.

Score quality in practice (moGC): Saying “we’re score-agnostic” helps, but there’s still no strong evidence here that better ASR confidence estimation actually improves the method (it’s more a promise of compatibility than a demonstrated win).

**Reviewer Scores:**

Reviewer ceGk: Their main weaknesses/questions (monotonicity gap, bigger LLMs, and auto-tuning) were all answered with concrete changes/experiments, so it’s reasonable they’d move from “weak accept” to a more comfortable accept.
Reviewer v9jL: They explicitly said an extra non-ASR study (speech translation correction) could increase their score, and the rebuttal adds exactly that plus fixes the theory gap, so a score bump is plausible.
Reviewer moGC: The authors did respond point-by-point (especially by switching to LTT and adding risk-control validation), but moGC’s review reads like a fundamental novelty + confidence-score skepticism, so the most likely change is a small nudge upward rather than a full reversal.

---

### Decision · Program_Chairs · 2026-01-26

Accept (Poster)